# CROSS-TIMESTEP: 3D DIFFUSION MODEL WITH TRANS-TEMPORAL MEMORY LSTM AND ADAPTIVE PRIORI DECODING STRATEGY FOR MEDICAL SEGMENTATION

**Shangqian Wu**[1], **Siyuan Shen**[1], **Yahan Li**[1], **Zhijian Huang**[1], **Ziyu Fan**[1], **Yuanpeng Zhang**[1], **Yi Wang**[2,*], **Lei Deng**[1,*]

[1]School of Computer Science and Engineering, Central South University, ChangSha, China
[2]Peking University People's Hospital
{wushangqian,leideng}@csu.edu.cn
[*]Corresponding author

## ABSTRACT

Diffusion models have recently demonstrated significant robustness in medical image segmentation, effectively accommodating variations across different imaging styles. However, their applications remain limited due to: (i) current successes being primarily confined to 2D segmentation tasks—we observe that diffusion models tend to collapse at the early stage when applied to 3D medical tasks; and (ii) the inherently isolated iteration along timesteps during training and inference. To tackle these limitations, we propose a novel framework named Cross-Timestep, which incorporates two key innovations: an Adaptive Priori Decoding Strategy (APDS) and a trans-temporal memory LSTM (tLSTM) mechanism. (i) The APDS provides prior guidance during the diffusion process by employing a Priori Decoder(PD) that focuses solely on the conditional branch, successfully stabilizing the reverse diffusion process. (ii) The tLSTM integrates convolution and linear layers into the LSTM gating structure, and enhances the memory cell mechanism to retain temporal state, explicitly preserving and propagating continuous temporal states across timesteps. Experimental results demonstrate that Cross-Timestep performs favorably on heterogeneous 3D medical datasets. Three experiments further analyze the collapse phenomenon in 3D medical diffusion models and validate that APDS effectively prevents initial-stage collapse without excessively constraining the model, while tLSTM facilitates the performance and scalability of diffusion models.

## 1 INTRODUCTION

Medical image segmentation—the precise delineation of anatomical structures—is essential for diagnosis and treatment planning (Zhang et al., 2025b; Luo et al., 2021a; 2022). Building robust models usually requires aggregating images across hospitals and scanners, which introduces large appearance variability and degrades the performance of conventional networks (Zaman et al., 2024; Luo et al., 2021b). Denoising Diffusion Probabilistic Models (DDPMs) are an attractive alternative because their coarse-to-fine denoising process encourages recovery of global structure before fine details, suggesting greater robustness to style variation (Salsi et al., 2025; Zhang et al., 2025a; Ding et al., 2024; Wu et al., 2025; Shuai et al., 2024).

Despite this promise, applying diffusion models to 3D volumetric segmentation remains challenging. We identify a failure mode we call "initial-stage collapse": when reverse sampling is started from very high-noise timesteps (near pure Gaussian noise), models adapted from 2D often produce incoherent outputs and fail to recover the target structure (see Fig. 1), additional details and visualizations are provided in Appendix A. The difficulty stems from the enlarged manifold of 3D volumes and the vanishingly weak structural cues present at extreme noise levels, while standard diffusion samplers lack mechanisms to accumulate evidence across timesteps.

To address these issues, we propose Cross-Timestep, a 3D diffusion framework that (i) uses an Adaptive Priori Decoding Strategy (APDS) to supply time-weighted structural priors from the conditional image, and (ii) equips the denoiser with an explicit cross-timestep memory via a transtemporal memory LSTM (tLSTM). APDS provides a coarse, image-driven scaffold that is strongest at early denoising steps and decays as the learned denoiser gains confidence; tLSTM persistently carries low-frequency structure, residual statistics, and uncertainty-aware saliency across steps so later timesteps refine rather than re-discover structure. We implement two concrete tLSTM instantiations (Conv-tLSTM and Linear-tGRU) and two extensions (SC-tLSTM and FFT-tLSTM) to demonstrate the approach's flexibility. Experiments on multi-center 3D datasets show Cross-Timestep prevents initial-stage collapse and improves segmentation robustness under severe domain shift. In summary, the contributions of this study are as follows:

- We introduce Cross-Timestep, pioneering the use of diffusion models for 3D medical segmentation tasks. This framework establishes a foundational strategy for diffusion-based approaches, offering a possible universal solution for future research and achieving robust performance on heterogeneous datasets.

- We propose the APDS to specifically address 'initial-stage collapse', providing stable structural guidance during high-noise stages without interfering with later refinement stages.

- We develop the tLSTM module, introducing a recurrent controller that carries cross-timestep state, turning local, myopic denoising into a temporally coherent trajectory.

- Implement tLSTM as a modular controller (Conv-tLSTM, Linear-tGRU) and demonstrate extensions (SC-tLSTM, FFT-tLSTM) that demonstrate substantial scalability and potential for further enhancements of diffusion models via these units.

## 2 RELATED WORK

Diffusion models have been adapted for discriminative tasks including image segmentation, with many 2D methods integrating conditional guidance, uncertainty modeling, or transformer-based context modules to improve boundaries and global consistency (Zaman et al., 2026; Ji & Chung, 2024; Wu et al., 2024; Tang et al., 2024). Work on 3D volumetric diffusion has focused mainly on generative synthesis and on engineering memory/efficiency solutions to handle large 3D tensors (Nie et al., 2025; Liu et al., 2025; Wang et al., 2024a; Chen et al., 2024), while some recent methods (e.g., Diff-UNet) fuse multistep outputs to reduce variance (Xing et al., 2025).

These prior approaches either (i) concentrate on generation and computational feasibility in 3D, or (ii) apply post-hoc fusion of independent per-step predictions. In contrast, Cross-Timestep targets two complementary problems simultaneously: reliable initialization from high-noise timesteps (via APDS) and explicit temporal features accumulation during sampling (via tLSTM). By integrating structured priors with a stateful denoiser, our framework addresses the root causes of initial-stage collapse rather than relying on post-hoc output fusion.

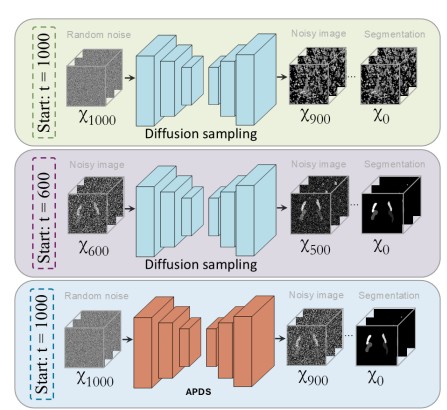

Figure 1: "Initial-stage collapse". For 3D medical data, the diffusion model will crash when sampling starts from the high-noise stage (equivalent to random noise), but it can correctly sample from the middle and low time steps. Introducing APDS enables correct sampling starting from random noise.

## 3 METHODOLOGY

In this section, we introduce our proposed Cross-Timestep method for stable 3D medical image segmentation using diffusion models (Fig. 2). We aim to make the reverse diffusion process both reliably initiable under extreme noise (via APDS) and temporally coherent across steps (via tLSTM).

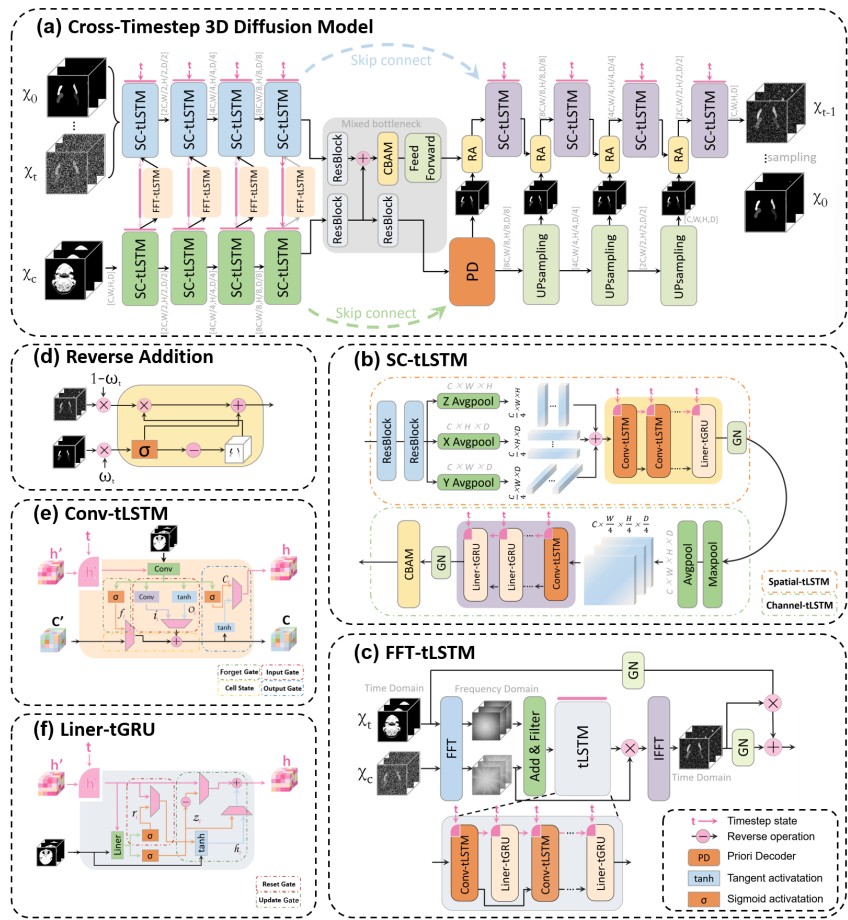

Figure 2: (a) The framework of the Cross-Timestep, we propose APDS and tLSTM to construct a stable diffusion architecture. (b) Time-weighted control RA, using the prior mask obtained from the PD to guide the main branch. (c) Detailed design of Conv-tLSTM, using convolution to improve the gating mechanism of LSTM and enhance its memory cell to remember the temporal state. (d) Detailed design of Linear-tGRU, using Linear combined with GRU to reduce resource requirements, compared with Conv-tLSTM. (e) Structure of SC-tLSTM, improving the traditional SC attention to adapt to the diffusion model. (f) Structure of FFT-tLSTM, transforming the time domain into the frequency domain for denoising.

APDS provides structural priors from the clean conditional image and injects them with a decaying time weight. tLSTM explicitly carries cross-timestep memory so evidence accumulates instead of being re-discovered.

## 3.1 ADAPTIVE PRIORI DECODING STRATEGY

As illustrated in Fig. 2a,b, the APDS mechanism operates exclusively on the conditional branch of the network. It takes the conditional image $X_c$, which is embedded with the current timestep information $t$. The deepest features from this are passed through a series of bottleneck blocks before entering a Prior Decoder (PD). The PD fuses these bottleneck features with multi-scale skip connections from the conditional encoder's path. This process yields a preliminary segmentation mask, $F_{prior}$, which serves as a robust approximation of the target structure $x_0$. According to the Reverse Addition(RA)(Fig. 2b) concept, $F_{prior}$ is reversed to guide the main branch $F_{main}$, thereby generating the refined feature map, $F_{refined}$:

$$F_{refined} = F_{main} \odot (1 - \sigma(F_{prior})), \tag{1}$$

Although there was no explicitly shown embedded time $t$ during this process, $X_c$ already incorporates the time embedding information, as it allows the APDS to be aware of the current diffusion

stage, enabling it to provide more global, structural guidance during high-noise timesteps and more detailed, refined guidance as the noise level decreases.

Further still, to prevent this strong prior from overly interfering with the main model's predictions, especially during the later, low-noise stages of diffusion, we introduce a time-weighted mechanism. The RA module is integrated at each upsampling stage of the main branch, and fuses them using a time-dependent weight, $\omega_t$. The fused feature map, $F_{fused}$, is computed as:

$$F_{fused} = (1 - \omega_t) \odot F_{refined} + \omega_t \odot F_{prior}. \tag{2}$$

The weight $\omega_t$ is designed to be high for large values of $t$ (initial high-noise stages) and diminish as $t$ approaches 0. This ensures that the guidance from the prior is strongest when the main branch is most unstable and gracefully recedes as the model's own predictive capabilities become more reliable, for details, please refer to Appendix B. The above design constitutes APDS, which can be seamlessly integrated into any 3D diffusion model to form a stable infrastructure.

## 3.2 CONV-TLSTM

The Conv-tLSTM module, illustrated in Fig. 2c, extends the classical LSTM to 3D volumetric data by replacing matrix multiplications with 3D convolutions in all gating operations. Unlike conventional LSTMs that only store abstract vector states, Conv-tLSTM maintains both the hidden state $h_t$ and the memory cell $C_t$ as full 3D tensors, thereby preserving spatial correlations across voxels.

To align with our cross-timestep memory design, we modify the hidden state update with a time-aware modulation. Specifically, the previous hidden state $h_{t-1}$ is first modulated by a timestep embedding $E_t$, producing $h'_t$, which allows the recurrent unit to remain aware of the denoising stage.

All three gates—input gate $i_t$, forget gate $f_t$, and output gate $o_t$—are computed using 3D convolutions:

$$i_t = Conv(W_{xi} * X_t + W_{hi} * h'_t + b_i), \ f_t = \sigma(W_{xf} * X_t + W_{hf} * h'_t + b_f), \tag{3}$$

$$o_t = \sigma(W_{xo} * X_t + W_{ho} * h'_t + b_o), \tag{4}$$

where $X_t$ is the noisy input at timestep $t$, and $*$ denotes 3D convolution.

Based on these gates, the candidate memory $\tilde{C}t$ is computed as:

$$\tilde{C}t = \tanh(W_{xc} * X_t + W_{hc} * h'_t + b_c), \tag{5}$$

and the cell state is updated as:

$$C_t = f_t \odot C_{t-1} + i_t \odot \tilde{C}_t. \tag{6}$$

Finally, the hidden state is updated as:

$$h_t = o_t \odot \tanh(C_t). \tag{7}$$

Intuitively, the cell state $C_t$ serves as the cross-timestep memory carrier, explicitly retaining: Low-frequency structural sketches ($\tilde{C}_t$ contributions), Residual noise statistics (filtered through $f_t$), Uncertainty-aware saliency cues (modulated by $i_t$ and $o_t$). Thus, Conv-tLSTM transforms the reverse diffusion process into a memory-guided trajectory where each denoising step refines, rather than re-discovers, structural evidence.

## 3.3 LINEAR-TGRU

While Conv-tLSTM excels at capturing spatial dependencies, its reliance on 3D convolutions incurs significant computational cost. To achieve a more lightweight yet effective recurrent design, we propose Linear-tGRU (Fig. 2d), which builds upon the Gated Recurrent Unit (GRU) architecture. Compared to LSTM, GRU merges the cell and hidden states into a single hidden representation, and fuses the forget and input gates into an update gate, reducing redundancy.

Similar to Conv-tLSTM, Linear-tGRU introduces timestep-aware modulation by embedding the current step $t$ into the hidden state $h_{t-1}$, yielding $h'_t$. Two gates are then computed:

$$r_t = \sigma(W_{xr}X_t + W_{hr}h'_t + b_r), \ z_t = \sigma(W_{xz}X_t + W_{hz}h'_t + b_z). \tag{8}$$

The reset gate $r_t$ controls how much past information is forgotten, while the update gate $z_t$ balances between retaining history and incorporating new evidence. The update of the candidate hidden state is provided in Appendix C.

By design, Linear-tGRU primarily retains global descriptors of the diffusion trajectory. Specifically, $z_t$ ensures persistent memory of coarse structures and residual statistics across timesteps, while $r_t$ selectively refreshes saliency regions when strong new evidence emerges. Compared with Conv-tLSTM, it sacrifices fine-grained spatial encoding for computational efficiency, making it suitable for long-horizon denoising.

## 3.4 SC-TLSTM

Building upon our foundational Conv-tLSTM and Linear-tGRU units, the Spatial-Channel Trans-temporal Memory LSTM (SC-tLSTM) module(Fig. 2e) adapts the conventional spatial-channel attention mechanism to be stateful and temporally aware.This module processes an input feature map through two recurrent branches—a spatial attention branch and a channel attention branch.The SC-tLSTM module creates a dynamic, stateful version of spatial-channel attention, using our recurrent units to learn what and where to focus throughout the entire reconstruction process.

### 3.4.1 SPATIAL ATTENTION BRANCH

we first generate compact feature descriptors from the input feature map $F \in \mathbb{R}^{C \times D \times H \times W}$. This is achieved by applying average pooling independently along the X, Y, and Z axes, thereby creating three distinct spatial summaries, $P_{xyz}$. These summaries are concatenated and fed into a recurrent block, primarily composed of 'Conv-tLSTM', which maintains a memory of spatial patterns across diffusion timesteps. The output is a 3D spatial attention map, $M_s$, computed as:

$$P_{xyz} = Concat(Pool_x(F), Pool_y(F), Pool_z(F)), \tag{9}$$
$$M_s = tLSTM(P_{xyz}), \tag{10}$$

### 3.4.2 CHANNEL ATTENTION BRANCH

Concurrently, to model the temporal evolution of inter-channel relationships, the module aggregates spatial information using both average-pooling and max-pooling operations across the spatial dimensions of the input feature map $F$. The resulting channel descriptors $P_{channel}$ are then processed by another recurrent block, which predominantly uses 'Linear-tGRU' . This branch tracks the evolution of channel importance over time, producing a channel attention map, $M_c$:

$$P_{channel} = Concat(AvgPool(F), MaxPool(F)), \tag{11}$$
$$M_c = tGRU(P_{channel}). \tag{12}$$

### 3.4.3 FEATURE REFINEMENT

The two attention maps are applied sequentially to the input feature map $F$ to adaptively refine it, the channel attention is applied first, followed by the spatial attention, yielding the final refined output, $F_{out}$:

$$F' = M_c \odot F, \tag{13}$$
$$F_{out} = M_s \odot F'. \tag{14}$$

## 3.5 FFT-TLSTM

The core principle of FFT-tLSTM, shown in Fig. 2f, is to leverage the frequency domain, where structural information and noise components are often more separable. The mechanism involves a sequence of transformations and stateful modulations. First, both the noisy input image $X_t$ and the conditional image $X_c$ are transformed from the spatial domain to the frequency domain using a 3D Fast Fourier Transform (FFT), $\mathcal{F}_t$ and $\mathcal{F}_c$:

$$\mathcal{F}_t = FFT(X_t), \quad \mathcal{F}_c = FFT(X_c), \tag{15}$$

The noisy and conditional spectra are then combined and filtered. This fused representation is processed by a stateful recurrent block. Similarly, the memory of time steps enables it to better grasp the scale of frequencies that belong to noise in the frequency domain space. The output is subsequently

modulated by the conditional spectrum $\mathcal{F}_c$, which acts as a gate to amplify relevant structural frequencies, obtained $\tilde{\mathcal{F}}$. This modulated spectrum is then transformed back to the spatial domain via an inverse FFT (iFFT), obtain the noisy image after frequency-domain denoising. Finally, the result is combined with the original noisy input through a residual connection, yielding final output, $X_{out}$.The entire process can be formulated as follows:

$$\tilde{\mathcal{F}} = tLSTM(Filter(\mathcal{F}_t + \mathcal{F}_c)) \odot \mathcal{F}_c, \tag{16}$$

$$X_{out} = iFFT(\tilde{\mathcal{F}}) + X_t. \tag{17}$$

### 3.6 DIFFUSION PROCESS

The diffusion process, whose overall network structure is depicted in Fig. 2a, consists of two primary stages: a fixed forward diffusion process and a learned reverse denoising process.The forward process progressively corrupts the ground truth segmentation mask $x_0$ over a sequence of $T$ timesteps by adding Gaussian noise. This is a fixed Markov chain defined as:

$$q(x_t|x_{t-1}) = \mathcal{N}(x_t; \sqrt{1-\beta_t}x_{t-1}, \beta_t \mathbf{I}). \tag{18}$$

where $\{\beta_t\}_{t=1}^T$ is a predefined variance schedule. As $t \to T$, the distribution of $x_T$ approaches a standard isotropic Gaussian distribution, $x_T \sim \mathcal{N}(0, \mathbf{I})$.

The reverse process is where our neural network learns to denoise these corrupted inputs. It is a learned Markov chain that begins with pure Gaussian noise $x_T$ and iteratively refines it to generate the final segmentation mask $x_0$. To achieve image-conditional segmentation, the reverse process is guided by the clean medical image, $X_c$. Our network, denoted as $\mathcal{M}_\theta$, is trained to predict the noise component $\epsilon$ from the noisy mask $x_t$ at a given timestep $t$, conditioned on $X_c$. The training objective follows the simplified loss function common in DDPMs:

$$\mathcal{L}_{simple} = \mathbb{E}_{t,x_0,\epsilon}\left[||\epsilon - \mathcal{M}_\theta(x_t, X_c, t)||^2\right]. \tag{19}$$

where $x_t = \sqrt{\bar{\alpha}_t}x_0 + \sqrt{1-\bar{\alpha}_t}\epsilon$ and $\epsilon \sim \mathcal{N}(0, \mathbf{I})$.

At inference time, the segmentation process starts by sampling a tensor of pure Gaussian noise, $x_T$. The model then iteratively applies the learned denoising function for $t = T, \ldots, 1$ to produce a progressively cleaner sample $x_{t-1}$ from $x_t$. This iterative refinement, guided at each step by the innovations within our architecture, can be formally expressed as:

$$\epsilon_\theta = \mathcal{D}_{\text{SC,APDS}}(\mathcal{E}_{\text{SC,FFT}}(x_t, t), X_c, t), \tag{20}$$

$$x_{t-1} = \frac{1}{\sqrt{\alpha_t}}\left(x_t - \frac{1-\alpha_t}{\sqrt{1-\bar{\alpha}_t}}\epsilon_\theta\right) + \sigma_t z. \tag{21}$$

Here, the sampling step utilizes the predicted noise $\epsilon_\theta$, which is generated by our complete network. The term $\mathcal{E}_{\text{SC,FFT}}$ represents the encoder, which leverages 'FFT-tLSTM' for noise resilience and 'SC-tLSTM' for stateful feature extraction. The term $\mathcal{D}_{\text{SC,APDS}}$ represents the decoder, which uses 'SC-tLSTM' blocks for reconstruction and is critically stabilized by the 'APDS' mechanism using the conditional image $X_c$. This process is repeated until the final, high-fidelity segmentation mask $x_0$ is produced. At each step we update a cross-timestep state $\mathcal{S}_t$ and condition the denoiser on $\mathcal{S}_t$, enabling evidence accumulation across steps:

$$\mathcal{S}_t = tLSTM(\mathcal{S}_{t+1}, \phi(x_t, X_c, t)) \tag{22}$$

## 4 EXPERIMENTS

### 4.1 EXPERIMENTAL SETUP

#### 4.1.1 DATASETS

To rigorously evaluate the robustness and generalization of our proposed model under domain shift, we utilized two multi-center nasopharyngeal carcinoma (NPC) segmentation datasets: LNCTVSeg (Luo et al., 2024; 2025) and OAseg (Wang et al., 2024b). LNCTVSeg comprises 440 computed tomography (CT) images from 262 patients across four medical institutions, focusing on lymph node clinical target volume (CTV) delineation. OAseg includes T1-weighted MRI scans for Gross Tumor Volume (GTV) segmentation collected from three distinct institutions: Center A (50 cases), Center B (50 cases), and Center C (60 cases). t-SNE visualization (Appendix D) confirmed significant stylistic variability across these datasets, providing an ideal testbed for assessing model performance under substantial domain shifts.

### 4.1.2 BASELINE METHODS

We compared our method against several state-of-the-art (SOTA) segmentation models, including TransBTS (Wang et al., 2021), SwinUNETR (Hatamizadeh et al.), UNETR (Hatamizadeh et al., 2022), 3DUXNET (Lee et al., 2022), nnFormer (Zhou et al., 2023), and Perspective+ (Hu et al., 2024). Detailed training parameters are described in Appendix E.

## 4.2 ADDRESSING 'INITIAL-STAGE COLLAPSE'

To empirically demonstrate the phenomenon of 'initial-stage collapse', we assessed the model's segmentation capability by initiating the reverse diffusion process from varying initial noise levels (timesteps ranging from low-noise (t=100) to high-noise (t=1000)). The average Dice scores were subsequently calculated as a function of these starting timesteps.

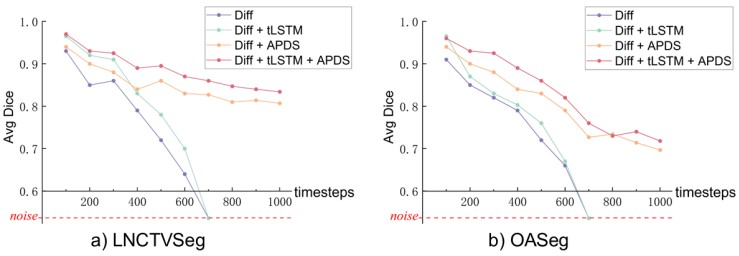

Figure 3: The average dice value obtained by performing reverse diffusion starting from noisy images at different time steps.

The results, depicted in Fig. 3 for both the LNCTVSeg and OAseg datasets, clearly illustrate this collapse. The baseline diffusion model ('Diff') and the variant enhanced solely with our temporal consistency module ('Diff + tLSTM') exhibited robust segmentation accuracy at low-to-moderate noise levels (t <600). However, their performance dramatically deteriorated as noise increased, collapsing entirely at timesteps beyond 700, where the input images were equivalent to random noise, resulting in incoherent predictions.

Conversely, models incorporating APDS ('Diff + APDS' and the full 'Diff + tLSTM + APDS' variant) exhibited remarkable stability across all noise levels. Although their performance slightly declined with increased noise, these models consistently maintained effective segmentation capabilities, even in high-noise regimes where conventional models failed. This result conclusively indicates that APDS provides essential structural guidance, preventing 'initial-stage collapse' and ensuring the reverse diffusion process remains on a viable solution trajectory from the outset. Consequently, APDS emerges as a universally applicable framework for enhancing stability in future 3D diffusion-based medical segmentation models.

## 4.3 MITIGATING APDS OVER-INTERFERENCE

A potential concern regarding the introduction of the Adaptive Priori Decoding Strategy (APDS) is its strong structural prior, which might inadvertently cause the model to overly depend on this guidance. Such dependence could degrade the model into a simpler refinement mechanism resembling a conventional U-Net. To investigate potential over-interference by APDS, we conducted an experiment to compare the performance between the main diffusion output ('Diff Out') and the APDS-generated prior ('APDS Out') throughout the reverse diffusion process.

Specifically, the reverse diffusion process commenced from timestep t=1000, and Dice scores were recorded for both 'Diff Out' and 'APDS Out' at each subsequent timestep. The results, depicted in Fig. 4, indicate that at initial high-noise stages, the 'Diff Out' initially struggles, relying heavily on the stable structural prior provided by APDS. However, as the timesteps decrease, the performance of 'Diff Out' rapidly improves, eventually surpassing 'APDS Out' and achieving significantly higher accuracy. This trend confirms that APDS effectively functions as intended—a supportive structural scaffold during unstable early stages—without constraining the model's learning capacity. Con-

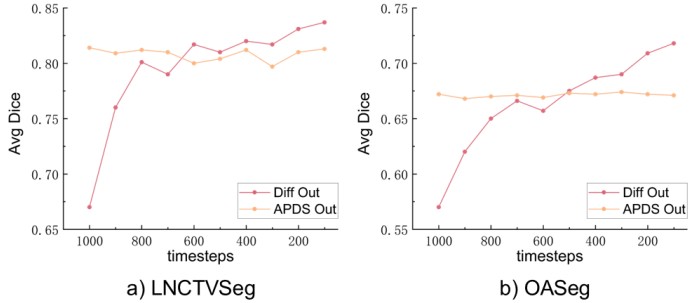

a) LNCTVSeg       b) OASeg

Figure 4: During a complete reverse diffusion process, the changes in the predicted results by the main denoising network ('Diff Out') and the Prior Decoder within APDS ('APDS Out').

sequently, our approach maintains the progressive refinement characteristic inherent to diffusion models, successfully mitigating concerns about APDS over-interference.

Table 1: Ablation study on tLSTM components.

| Modules | | | | LNCTVSeg | | | OASeg | | |
|---|---|---|---|---|---|---|---|---|---|
| LSTM | Conv-LSTM | Linear-GRU | t-cell | Dice ↑ | IoU ↑ | HD95 ↓ | Dice ↑ | IoU ↑ | HD95 ↓ |
| ✓ | | | | 79.5 | 66.6 | 6.93 | 69.8 | 53.7 | 8.94 |
| ✓ | ✓ | | | 81.1 | 68.2 | 5.48 | 70.6 | 54.6 | 8.89 |
| ✓ | | ✓ | | 79.2 | 65.6 | 7.12 | 69.1 | 52.8 | 10.11 |
| ✓ | ✓ | ✓ | | 82.5 | 70.2 | 3.81 | 71.7 | 56.0 | 7.49 |
| ✓ | ✓ | ✓ | ✓ | **83.7** | **74.2** | **2.44** | **72.8** | **65.4** | **6.24** |

## 4.4 TRANS-TEMPORAL MEMORY

The tLSTM module aims to ensure temporal consistency within the reverse diffusion process by preserving stateful memory across timesteps. To qualitatively validate this capability, we visualized the model's feature heatmap across different stages of the denoising trajectory for a representative sample, as illustrated in Fig. 5 (details provided in Appendix F). Initially, at the high-noise stage (t=999), the model's attention is diffuse, emphasizing broad structural reconstruction. As denoising progresses, the tLSTM leverages its temporal memory, refining attention patterns and increasingly focusing on the target structure. By timestep t=799, the model starts developing a more defined target-region focus, which progressively sharpens at timesteps t=399 and t=100. This visualization clearly demonstrates the evolving stateful attention driven by tLSTM's temporal memory.

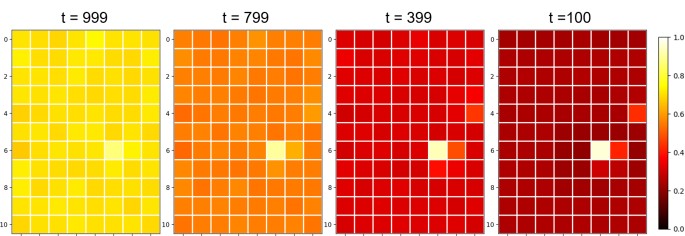

Figure 5: An example of the heat map during the reverse diffusion.

To evaluate the efficacy of the tLSTM module, we conducted a comprehensive ablation study quantifying its performance contributions (Table 1). Beginning with a baseline model devoid of temporal modules, the introduction of a standard LSTM yielded notable performance improvements. Replacing standard LSTM with our specialized 'Conv-LSTM',which integrates 3D convolutional operations with LSTM gates, and 'Linear-GRU', which combines linear layers with the GRU structure, specifically designed for 3D imaging, further enhanced performance. The optimal performance was attained by integrating our proposed 't-cell' innovation. These findings underscore the importance of explicit temporal awareness in achieving maximum stability and accuracy.

Moreover, this analysis validates the versatility of 'Conv-tLSTM' and 'Linear-tGRU' units, demonstrating their fundamental role as adaptable attention modules within diffusion models. Their recombination potential offers extensive opportunities for future architectural innovations and advancements.

## 4.5 COMPARISON WITH STATE-OF-THE-ART METHODS

We conducted a comprehensive comparison between our proposed model and several state-of-the-art (SOTA) segmentation methods on the LNCTVSeg and OAseg datasets. As detailed in Table 2, our model consistently outperformed existing approaches across all evaluated metrics. Specifically, our method achieved the highest Dice and IoU scores, demonstrating superior accuracy in volumetric overlap with ground truth segmentations. Additionally, our model exhibited the lowest HD95 scores, reflecting enhanced precision in segmentation boundary delineation compared to other leading methods. Visual comparisons provided in Appendix G further illustrate these advantages. These results underscore the robustness and effectiveness of our proposed architecture, particularly when addressing significant stylistic variability across multi-center medical imaging datasets.

Table 2: Comparison with state-of-the-art methods.

| Method | LNCTVSeg | | | OASeg | | |
|---|---|---|---|---|---|---|
| | Dice↑ | IoU↑ | HD95↓ | Dice↑ | IoU↑ | HD95↓ |
| TransBTS | 78.6 | 68.7 | 5.14 | 66.3 | 55.4 | 12.7 |
| SwinUNETR | 80.7 | 71.1 | 4.14 | 65.9 | 58.3 | 9.50 |
| UNETR | 78.1 | 69.1 | 6.87 | 67.1 | 59.2 | 9.04 |
| nnFormer | 80.3 | 71.5 | 4.31 | 68.4 | 62.4 | 7.76 |
| 3DUXNET | 81.6 | 72.9 | 3.52 | 68.9 | 61.3 | 7.93 |
| Perspective+ | 82.4 | 73.6 | 3.27 | 69.6 | 62.8 | 7.09 |
| Diff-UNet | 81.7 | 72.2 | 3.91 | 71.5 | 64.2 | 6.88 |
| **Ours** | **83.7** | **74.2** | **2.44** | **72.8** | **65.4** | **6.24** |

## 4.6 QUANTITATIVE COMPARISON OF COMPUTATIONAL COST

To more rigorously quantify the computational efficiency of the proposed Cross-Timestep framework, we conducted a comparative analysis against several representative 3D segmentation architectures—namely UX-Net, Swin-UNETR, nnFormer, and the diffusion-based Diff-UNet—on the LNCTVSeg dataset. Four complementary indicators were assessed to capture overall computational cost: training time, inference latency, GFLOPS, and GPU memory consumption. All models were evaluated under consistent experimental conditions, and the complete protocol is documented in Appendix H.

Table 3: Ablation study on APDS, SC, and FFT modules.

| Modules | | | LNCTVSeg | | | OASeg | | |
|---|---|---|---|---|---|---|---|---|
| APDS | SC | FFT | Dice↑ | IoU↑ | HD95↓ | Dice↑ | IoU↑ | HD95↓ |
| ✓ | | | 77.3 | 63.0 | 8.51 | 67.5 | 50.9 | 11.53 |
| ✓ | ✓ | | 82.1 | 69.6 | 4.32 | 71.3 | 55.4 | 7.83 |
| ✓ | | ✓ | 81.8 | 69.2 | 4.85 | 70.9 | 54.9 | 8.14 |
| ✓ | ✓ | ✓ | **83.7** | **74.2** | **2.44** | **72.8** | **65.4** | **6.24** |

As summarized in Table 4, the proposed method exhibits a balanced efficiency profile in comparison with both transformer-based and diffusion-based baselines. Notably, Cross-Timestep requires substantially less training time than large transformer architectures such as SwinUNETR and nnFormer, while maintaining a moderate memory footprint and competitive runtime. Moreover, relative to the diffusion-based Diff-UNet—which is optimized for lightweight computation—our method demonstrates comparable resource usage while offering markedly improved segmentation accuracy. These results collectively indicate that Cross-Timestep achieves a favorable trade-off between computational efficiency and predictive performance, strengthening its practicality for real-world clinical deployment.

## 4.7 ABLATION STUDY

To quantify the contribution of individual modules within our architecture, we performed an ablation study, as summarized in Table 3. The baseline model included the core diffusion framework and the essential Adaptive Priori Decoding Strategy (APDS) to address 'initial-stage collapse'. Introducing either the SC-tLSTM (SC) or FFT-tLSTM (FFT) modules individually significantly improved performance on both datasets, demonstrating the efficacy of spatio-channel temporal memory and

Table 4: Detailed computational cost including training/inference time, GPU memory usage, and GFLOPS

| Method | Training Time (h) | Inference Time (s) | GFLOPS | GPU Mem (MiB) |
|---|---|---|---|---|
| UX-Net | 33.6 | 0.11 | 631.9 | 14722 |
| SwinUNETR | 45.3 | 0.09 | 328.8 | 16364 |
| nnFormer | 80.4 | **0.03** | **235.3** | 11626 |
| Diff-UNet | **29.2** | 0.12 | 981.1 | **10834** |
| **Ours** | 33.4 | 0.17 | 453.5 | 15152 |

frequency-domain noise resilience enhancements. These findings highlight the versatility and potential of Conv-tLSTM and Linear-tGRU modules as foundational components that can be effectively leveraged to further extend and refine diffusion-based segmentation models.

## 5 CONCLUSION

In this study, we identified and addressed a critical challenge—'initial-stage collapse'—that significantly constrains the applicability of diffusion models in 3D medical segmentation tasks. We introduced Cross-Timestep, a versatile 3D diffusion framework integrating APDS and tLSTM. These components collectively provide structural stability during early diffusion phases and maintain temporal consistency across timesteps. Experiments conducted on multi-center datasets demonstrated that Cross-Timestep effectively resolves 'initial-stage collapse' and achieves state-of-the-art segmentation performance. Our framework provides a robust foundation for advancing 3D diffusion-based medical segmentation methodologies and illustrates considerable potential for integration with other sophisticated architectures.

**REPRODUCIBILITY** Our code and data are made publicly available at https://github.com/Wushangqian404/Cross-Timestep

**ACKNOWLEDGMENTS.** This research was supported by the National Natural Science Foundation of China (Grant Nos. U23A20321 and 62272490); the Natural Science Foundation of Hunan Province of China (Grant No. 2025JJ20062); the National Natural Science Foundation of China (Grant No. 82471964).

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

## SUPPLEMENTARY MATERIAL

### ETHICS STATEMENT

This work adheres to the ICLR Code of Ethics. In this study, no human subjects or animal experimentation was involved. All datasets used were sourced in compliance with relevant usage guidelines, ensuring no violation of privacy. We have taken care to avoid any biases or discriminatory outcomes in our research process. No personally identifiable information was used, and no experiments were conducted that could raise privacy or security concerns. We are committed to maintaining transparency and integrity throughout the research process.

### REPRODUCIBILITY STATEMENT

We have made every effort to ensure that the results presented in this paper are reproducible. The experimental setup, including training steps, model configurations, and hardware details, is described in detail in the paper.

Additionally, all datasets used in this paper are publicly available resources(https://anonymous.4open.science/status/Cross-Timestep-0D2A), ensuring the consistency and reproducibility of the evaluation results.

We believe these measures will enable other researchers to reproduce our work and further advance the field.

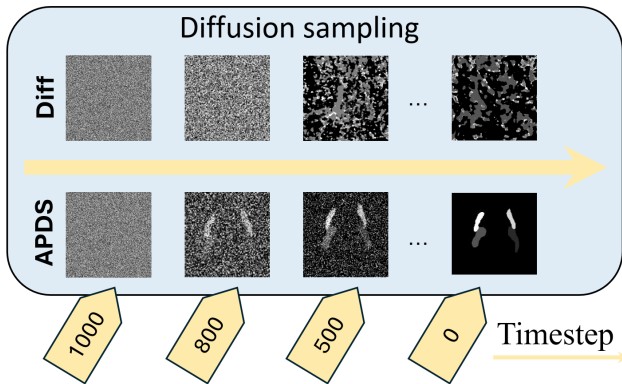

Figure 6: Visualization of the reverse diffusion process with and without APDS.

## LLM USAGE

Large Language Models (LLMs) were used to aid in the writing and polishing of the manuscript. Specifically, we used an LLM to assist in refining the language, improving readability, and ensuring clarity in various sections of the paper. The model helped with tasks such as sentence rephrasing, grammar checking, and enhancing the overall flow of the text.

It is important to note that the LLM was not involved in the ideation, research methodology, or experimental design. All research concepts, ideas, and analyses were developed and conducted by the authors. The contributions of the LLM were solely focused on improving the linguistic quality of the paper, with no involvement in the scientific content or data analysis.

The authors take full responsibility for the content of the manuscript, including any text generated or polished by the LLM. We have ensured that the LLM-generated text adheres to ethical guidelines and does not contribute to plagiarism or scientific misconduct.

## APPENDIX A

To further illustrate the stabilizing effect of APDS, Fig. 6 presents a visualization of the reverse diffusion process with and without APDS (APDS and Diff). We show representative timesteps ($t = 1000, 800, 500$) and the final prediction. Without APDS, the model produces unstable, structureless patterns at high-noise timesteps, reflecting the initial-stage collapse phenomenon. In contrast, APDS injects explicit prior guidance at early timesteps, allowing coarse anatomical structures to gradually emerge and resulting in a stable denoising trajectory throughout the diffusion process.

## APPENDIX B

The Adaptive Prior-guidance Denoising Schema (APDS) introduces a dynamic weighting factor, $\omega_t$, to modulate the influence of the external prior throughout the reverse diffusion process. The core design principle is to apply strong guidance during the initial, high-noise stages of generation (when $t$ is large) and to progressively reduce this guidance as the model's own generative capacity becomes more reliable in later, low-noise stages (as $t \to 0$), as illustrated in Fig. 7. This ensures stability without sacrificing the fine details learned by the model.

To provide explicit control over the maximum guidance strength, we introduce a hyperparameter, $\alpha \in [0, 1]$, which acts as a scaling factor or threshold. This allows for fine-tuning the upper limit of the prior's influence. For instance, a value of $\alpha = 0.5$ would cap the maximum weight at 50%. The weight $\omega_t$ is defined as a function of the normalized time $t_{\text{normalized}} \in [0, 1]$, where $t_{\text{normalized}} = t/T$ for a total of $T$ timesteps. The formulation is given by:

$$\omega_t = \alpha \cdot \exp(-5.0 \cdot (1 - t_{\text{normalized}})) \cdot (1 - \exp(-10.0 \cdot t_{\text{normalized}})) \tag{23}$$

To understand the behavior of $\omega_t$, we can analyze its constituent terms: Primary Decay Term ($1 - \exp(-10.0 \cdot t_{\text{normalized}})$): This component ensures that the guidance is nullified at the very end of

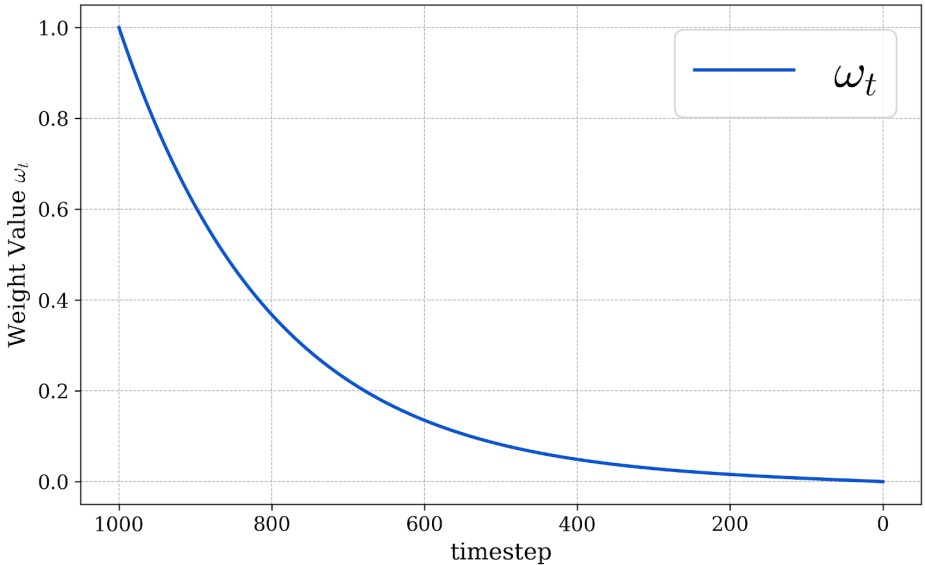

Figure 7: "Initial-stage collapse".

the diffusion process. As $t_{\text{normalized}} \to 0$, the term $\exp(-10.0 \cdot t_{\text{normalized}}) \to 1$, which causes the entire expression to approach 0. This is critical for allowing the model to generate high-fidelity details without being constrained by the prior. For $t_{\text{normalized}} > 0$, this term rapidly increases towards 1, effectively "activating" the guidance. Shaping Term ($\exp(-5.0 \cdot (1 - t_{\text{normalized}}))$): This term sculpts the decay curve. At the beginning of the process, when $t_{\text{normalized}} \to 1$, this term approaches $\exp(0) = 1$. Consequently, the overall weight $\omega_t$ approaches its predefined maximum value of $\alpha$. As $t_{\text{normalized}}$ decreases, this term introduces a smooth, exponential decay, ensuring a graceful transition from a high-guidance regime to a low-guidance one, rather than an abrupt cutoff.

The combined effect of these terms is a weighting curve that starts at 0, rises to a peak value of approximately $\alpha$ in the early-to-mid stages, and then decays back to 0. The plot of $\omega_t$ versus the normalized time (from 1 down to 0) visually confirms this behavior. This carefully designed weighting function is integral to the stability and efficacy of the APDS framework, providing robust guidance where needed while preserving the generative autonomy of the underlying 3D diffusion model.

APPENDIX C

The Conv-tLSTM module is the core recurrent component of our architecture, designed to process 3D volumetric feature maps across the denoising trajectory. It extends the standard LSTM by integrating 3D convolutional operations directly into its gating units, enabling it to effectively learn and propagate spatio-temporal information. This appendix provides a detailed breakdown of the mathematical operations within each gate and state update of the module.

In this module, all states ($C_t$, $h_t$) and inputs ($X_t$) are 3D tensors. The '$*$' operator denotes a 3D convolution, and $\odot$ denotes the Hadamard (element-wise) product. The previous hidden state $h_{t-1}$ is first modulated by a learned time embedding $E_t$ to produce a time-aware hidden state $h'_t$, which is then used in the gate computations.

The module's behavior is governed by three primary gating units that regulate the flow of information.

Input Gate ($i_t$): This gate determines which new information from the input $X_t$ will be stored in the cell state. It computes an update mask by combining the current input and the time-aware previous hidden state. Its computation is as follows:

$$i_t = \sigma(W_{xi} * X_t + W_{hi} * h'_t + b_i) \tag{24}$$

where $X_t$ is the current input, $W$ terms are 3D convolutional kernels, $b_i$ is the bias term, and $\sigma$ is the sigmoid activation function.

Forget Gate ($f_t$): This gate is responsible for deciding which information to discard from the previous cell state, $C_{t-1}$. This is critical for the model to forget irrelevant features from earlier stages of the long denoising trajectory. Its computation is:

$$f_t = \sigma(W_{xf} * X_t + W_{hf} * h'_t + b_f) \tag{25}$$

Output Gate ($o_t$): This gate controls which information from the updated cell state will be used to compute the new hidden state. It acts as a filter on the cell's memory to produce the module's output for the current timestep. It is computed as:

$$o_t = \sigma(W_{xo} * X_t + W_{ho} * h'_t + b_o) \tag{26}$$

Based on the outputs of these gates, the cell and hidden states are updated for the current timestep $t$.

First, a candidate cell state, $\tilde{C}_t$, is computed from the current input and previous hidden state. This represents the new information that *could* be added to the cell's memory.

$$\tilde{C}_t = \tanh(W_{xc} * X_t + W_{hc} * h'_t + b_c) \tag{27}$$

Next, the new cell state, $C_t$, is produced by combining the previous cell state, $C_{t-1}$, with the candidate state. The forget gate determines what to discard from $C_{t-1}$, while the input gate determines what to add from $\tilde{C}_t$.

$$C_t = f_t \odot C_{t-1} + i_t \odot \tilde{C}_t \tag{28}$$

Finally, the new hidden state, $h_t$, is determined by passing the updated cell state through a $\tanh$ function and then filtering it with the output gate.

$$h_t = o_t \odot \tanh(C_t) \tag{29}$$

Through this sequence of operations, the Conv-tLSTM module can selectively update its memory over time, preserving relevant spatial features while discarding irrelevant ones, making it a powerful tool for sequential generative processes.

The Linear-tGRU module serves as a Domputationally efficient recurrent component within our framework, designed to balance performance with operational cost. It adapts the Gated Recurrent Unit (GRU) architecture, replacing computationally intensive 3D convolutions with standard linear transformations. This makes it well-suited for processing feature vectors where full spatial convolution is not required. This appendix details the mathematical formulations of its gating units and state update process.

In this module, the '·' operator denotes matrix multiplication, and $\odot$ represents the Hadamard (element-wise) product. As with the Conv-tLSTM, the previous hidden state $h_{t-1}$ is modulated by a time embedding $E_t$ to create a time-aware hidden state $h'_t$ before being used in calculations.

The Linear-tGRU simplifies the information flow using two primary gating units:

Reset Gate ($r_t$): This gate determines how much of the previous hidden state should be forgotten when computing the new candidate hidden state. This allows the model to effectively discard irrelevant past information that is not pertinent to the current input. Its computation is:

$$r_t = \sigma(W_{xr} \cdot X_t + W_{hr} \cdot h'_t + b_r) \tag{30}$$

where $W$ terms are weight matrices, $b_r$ is the bias term, and $\sigma$ is the sigmoid function.

Update Gate ($z_t$): This gate functions as a combination of the forget and input gates in an LSTM. It decides how much of the previous hidden state to retain and how much of the new candidate state to incorporate, controlling the flow of information into the final hidden state. It is computed as:

$$z_t = \sigma(W_{xz} \cdot X_t + W_{hz} \cdot h'_t + b_z) \tag{31}$$

The final hidden state is updated in a two-step process using the outputs from the gating units.

First, a candidate hidden state, $\tilde{h}_t$, is computed. The reset gate ($r_t$) modulates the influence of the previous state ($h'_t$), allowing the model to focus on generating new information from the current input $X_t$.

$$\tilde{h}_t = \tanh(W_{xh} \cdot X_t + W_{hh} \cdot (r_t \odot h'_t) + b_h) \tag{32}$$

Finally, the new hidden state, $h_t$, is determined by a linear interpolation between the previous hidden state $h_{t-1}$ and the candidate hidden state $\tilde{h}_t$. The update gate $z_t$ controls this interpolation, balancing the information to be kept from the past with the new information to be added.

$$h_t = (1 - z_t) \odot h_{t-1} + z_t \odot \tilde{h}_t \tag{33}$$

This streamlined architecture allows the Linear-tGRU to efficiently update its state, capturing temporal dependencies without the overhead of maintaining a separate cell state, thus offering a powerful yet lightweight recurrent building block.

APPENDIX D

To visually substantiate the claim of significant domain shifts across the evaluation datasets, we employed the t-Distributed Stochastic Neighbor Embedding (t-SNE) technique. t-SNE is a non-linear dimensionality reduction method particularly well-suited for visualizing the structure of high-dimensional data in a low-dimensional space (e.g., a 2D plot). For this analysis, we extracted deep features from a representative set of images from each source institution and projected them into a 2D space. The resulting plots vividly illustrate the data distribution and inter-institutional variability.

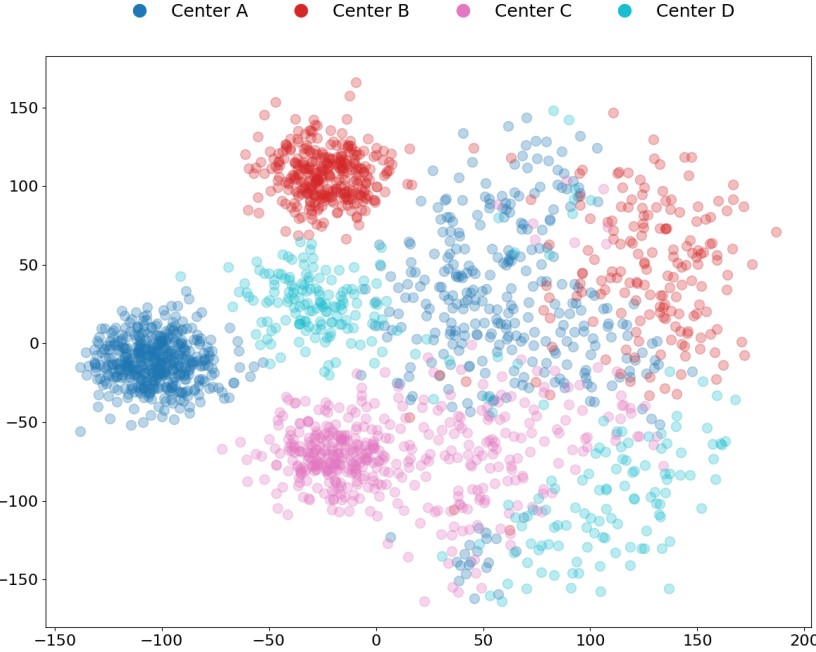

Figure 8: t-SNE visualization of the LNCTVSeg dataset. Data points are colored according to their source institution. The formation of separate clusters confirms the multi-center heterogeneity of the data.

Fig. 8 displays the t-SNE plot for the LNCTVSeg dataset. The data points, representing CT images from four different medical institutions, also form distinct groups. This visualization confirms that despite all being CT scans for NPC, there are notable stylistic and technical differences between the data from each source.

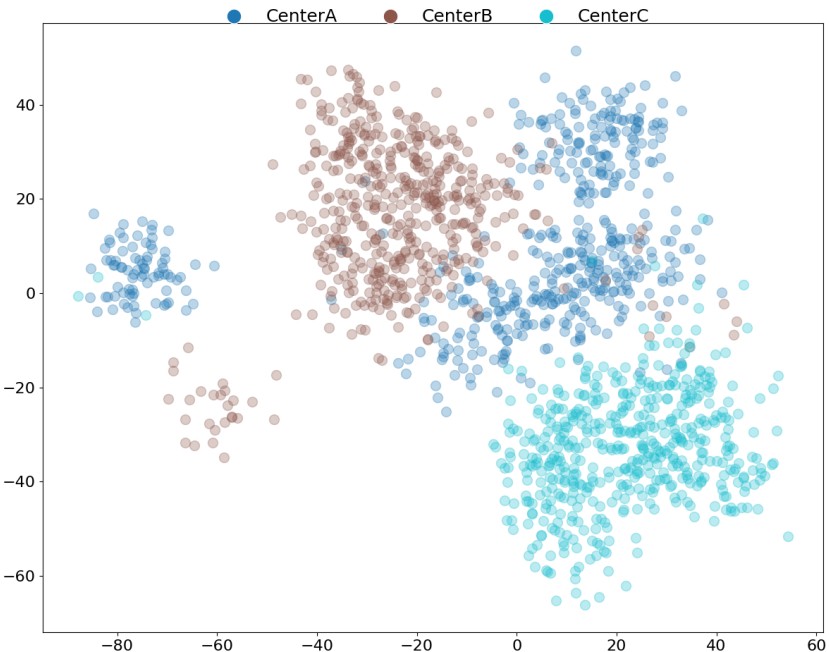

Figure 9: t-SNE visualization of the OAseg dataset. Data points are colored according to their center of origin (Center A, B, and C). The distinct clustering demonstrates significant domain shift across the three institutions.

As illustrated in Fig. 9, the t-SNE visualization of the OAseg dataset reveals three semi-distinct clusters, each corresponding to one of the contributing medical centers (Center A, Center B, and Center C). The clear separation between these clusters indicates substantial differences in the underlying data distributions. These variations likely arise from differences in MRI scanners, imaging protocols, and patient populations, presenting a significant domain adaptation challenge.

Collectively, these visualizations provide compelling evidence of the domain heterogeneity present both within each multi-center dataset and between the two datasets (CT vs. MRI). This confirms that our chosen datasets provide a rigorous and realistic testbed for evaluating the robustness and domain generalization capabilities of our proposed model.

## APPENDIX E

To ensure a fair and reproducible comparison, all experiments were conducted using a consistent training and evaluation framework. This appendix details the hardware, software, and hyperparameters used for training our proposed model and the baseline methods.

Hardware: All models were trained on a server equipped with four NVIDIA A6000 graphics cards, each with 48 GB of VRAM. Software: The implementation was based on the PyTorch deep learning framework (version 1.13). The training environment utilized Python 3.8, CUDA 11.7, and cuDNN 8.5 for optimized performance.

The following hyperparameters were used for training our model and for re-implementing the baseline models to ensure a fair comparison: Optimizer: We used the AdamW optimizer for all training processes, as it is well-suited for both CNN and Transformer-based architectures. Learning Rate: The initial learning rate was set to $5 \times 10^{-5}$. Learning Rate Scheduler: A cosine annealing scheduler was employed to adjust the learning rate over the course of training. A warm-up period of 50 epochs was used at the beginning of training, during which the learning rate was gradually increased to its initial value. Loss Function: A compound loss function, combining Dice Loss and Cross-Entropy Loss, was used to supervise the segmentation. The final loss was computed as $\mathcal{L} = \mathcal{L}_{\text{Dice}} + \mathcal{L}_{\text{CE}}$.

This combination provides a balance between region-based and pixel-wise accuracy. Epochs and Batch Size: All models were trained for a total of 200 epochs to ensure full convergence. Due to the memory constraints of 3D data, a batch size of 4 was used (distributed as 1 samples per GPU). Input Patch Size: During training, we extracted random patches of size $96 \times 96 \times 96$ from the full-resolution images to serve as inputs to the models.

To improve model robustness and prevent overfitting, we applied a series of on-the-fly data augmentation techniques to the input patches during training. These augmentations include: Random rotation in the range of [-30°, 30°], Random scaling with a factor between 0.7 and 1.4, Random elastic deformation, Random gamma correction with a gamma value between 0.7 and 1.5, Random horizontal and vertical flipping (mirroring).

For the baseline models (TransBTS, SwinUNETR, etc.), we utilized their officially available code-bases and default configurations where possible, and we carefully reproduced them within our framework using the same training strategy described above to maintain consistency across all comparisons.

APPENDIX F

To qualitatively evaluate the tLSTM module's ability to maintain stateful memory across the reverse diffusion process, we visualized the evolution of the model's internal feature representations at different timesteps. The feature heatmaps shown in Fig. 4 were generated from the output feature maps of the final tLSTM module in our model's architecture. The process began by selecting a representative sample from the test set and running the full reverse diffusion trajectory. We captured the output 3D feature tensor from t=999 to t=100. To create a single-channel heatmap from this multi-channel tensor, we computed the mean activation across the channel dimension, which summarizes the model's spatial focus at that instant. Finally, the resulting 3D heatmap was upsampled to the original image resolution and overlaid on the corresponding axial slice of the input image using a "jet" colormap, where red indicates high activation and blue indicates low activation.

The sequence of heatmaps in Fig. 10 provides a clear visual narrative of the tLSTM's temporal dynamics. Initially, at the high-noise stage, the model's attention is diffuse, with the heatmap showing broad, low-intensity activations that reflect the initial uncertainty and a focus on reconstructing the general anatomy. As the denoising process continues, the tLSTM begins to leverage its memory, causing the heatmap to concentrate as background activations diminish and the focus on the potential target region intensifies. This localization becomes significantly more pronounced, where the model's attention is clearly centered on the target structure. By the final refinement stage, the heatmap is sharp and precisely aligned with the target's boundaries, with background noise almost entirely suppressed. In summary, this progressive sharpening from a diffuse to a highly localized focus provides strong qualitative evidence that the tLSTM module effectively maintains and utilizes stateful temporal memory, a mechanism crucial for ensuring coherence and accuracy throughout the multi-step reverse diffusion process.

APPENDIX G

Fig. 11 presents qualitative segmentation results on both the LNCTVSeg (CT) and OAseg (MRI) datasets. To provide a comprehensive visual comparison, we benchmark our method against several strong and widely adopted 3D medical segmentation baselines, including SwinUNETR, nnFormer, 3DUXNET, Perspective+, and the diffusion-based model Diff-UNet.

Across both datasets, competing methods frequently show issues such as incomplete tumor coverage, over-segmentation into surrounding tissues, or blurred and irregular boundaries—particularly under strong domain shifts in the multi-center MRI data (OAseg). In contrast, our method consistently produces segmentations that exhibit more accurate localization, cleaner boundaries, and better volumetric alignment with the ground truth.

On the LNCTVSeg dataset, our model more faithfully captures the irregular lymph-node CTV shapes, while the baseline models often miss small subregions or introduce spurious fragments. On the OAseg dataset, where significant inter-center variability exists, our model maintains robust performance and yields cleaner GTV predictions, demonstrating strong generalizability.

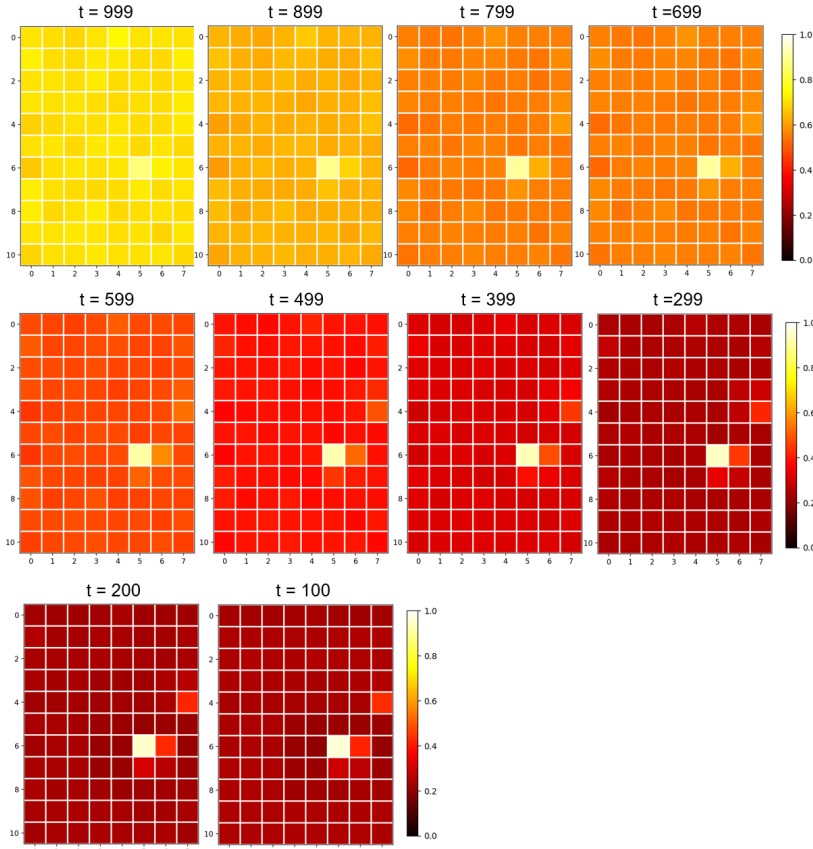

Figure 10: Visualization of feature heatmaps at different denoising timesteps. The progressive focusing of the heatmap demonstrates the tLSTM module's ability to leverage temporal memory to refine its attention on the target structure.

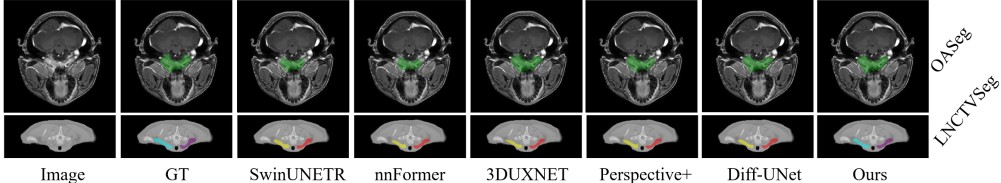

Figure 11: Visual comparison on representative cases from the LNCTVSeg and OAseg datasets.

These visual comparisons validate the quantitative results reported in the main paper, confirming that our approach not only achieves state-of-the-art numerical performance but also delivers more reliable and clinically meaningful segmentation outputs.

## APPENDIX H

To further evaluate the computational efficiency of the proposed framework, we benchmarked our model against several representative SOTA segmentation models on the LNCTVSeg dataset. All experiments were conducted under strictly controlled and consistent settings: the input voxel resolution was uniformly resampled to 96×96×96, the batch size was fixed at 1, and all models were trained and tested using the same GPU environment. For each method, we measured four key indicators—training time, inference time, GFLOPS, and GPU memory consumption—to provide a comprehensive assessment of computational cost.

As summarized in Table 4, the proposed model demonstrates a balanced computational profile. Although it is not the fastest model in inference nor the most memory-efficient, it achieves competitive performance while maintaining a moderate resource demand. In particular, our model requires substantially less training time than transformer-based architectures. These results suggest that the proposed model attains an effective trade-off between segmentation performance and computational efficiency.

APPENDIX I

Diffusion models inherently involve a trade-off between computational cost and reconstruction quality. Using fewer diffusion steps reduces runtime and memory consumption but may lead to degraded performance, while more steps typically improve output quality at the cost of higher computation. Following commonly adopted settings (500–1000 steps), we evaluated our framework on the LNCTVSeg dataset using 300, 500, and 1000 diffusion steps. The results are summarized in Table 5, showing that insufficient steps (e.g., 300) lead to notable performance drops, whereas 1000 steps yield the best overall accuracy.

Table 5: Performance under different diffusion steps on the LNCTVSeg dataset.

| Steps | Dice↑ | IoU↑ | HD95↓ |
|-------|-------|------|-------|
| 300 | 79.3 | 68.9 | 4.86 |
| 500 | 82.8 | 73.0 | 3.39 |
| 1000 | **83.7** | **74.2** | **2.44** |

