# OpenReview forum: "Cross-Timestep: 3D Diffusion Model with Trans-temporal Memory LSTM and Adaptive Priori Decoding Strategy for Medical Segmentation"
_ICLR.cc/2026/Conference — ICLR 2026 Poster_

### Official Review · Reviewer_1Gpu · 2025-10-27

**Soundness:** 2
**Presentation:** 2
**Contribution:** 2
**Rating:** 4
**Confidence:** 4

**Summary:**

The paper proposes a 3D diffusion framework for medical image segmentation that addresses the instability of conventional diffusion models when applied to volumetric data. It introduces two key components: the Adaptive Priori Decoding Strategy (APDS), which provides structural guidance to prevent early-stage collapse during high-noise timesteps, and the trans-temporal memory LSTM (tLSTM), which preserves temporal information across diffusion steps for consistent refinement. Together, these innovations stabilize the reverse diffusion process and improve segmentation accuracy across heterogeneous datasets, achieving state-of-the-art results and demonstrating strong robustness under domain shifts.

**Strengths:**

Enhanced temporal coherence: The trans-temporal memory LSTM (tLSTM) explicitly retains and propagates structural and contextual information across diffusion timesteps, turning each step into a coherent continuation rather than an independent reconstruction.

Improved stability under high noise: By combining tLSTM with the Adaptive Priori Decoding Strategy (APDS), the model accumulates temporal evidence effectively, preventing early-stage collapse and ensuring reliable recovery from highly noisy initial states.

**Weaknesses:**

What is the reason for the initial-stage collapse? Why are 2D models utilized in a 3D scenario?
Some methods, such as Diff-UNet, which use 3D diffusion models, can avoid the initial-stage collapse — is that correct?
Sometimes, the authors describe their methods in a confusing and complex way. For example, what does “explicit temporal evidence accumulation” mean?
There are no diffusion-based methods included in the comparison experiments — why?

**Questions:**

N/A

---

> ### Author Response · Authors · 2025-11-21
>
> We sincerely thank Reviewer for the detailed and insightful comments, which helped us improve both the technical clarity and empirical evaluation of our work. Below we address each concern point-by-point.
>
> ## **Q1. What is the exact cause of the initial-stage collapse?**
>
> Thank you for raising this important question. The *initial-stage collapse* is a practical instability we repeatedly observed when extending diffusion models from 2D medical images to full 3D volumes. We believe the issue arises from two major factors:
>
> 1. **Inter-slice dependency and high structural complexity** in 3D data significantly increase modeling difficulty compared to independent 2D slices.
> 2. **Directly applying 2D diffusion architectures to 3D** proportionally increases noise accumulation and gradient instability, often leading to collapse at early reverse-diffusion timesteps.
>
> While Diff-UNet briefly mentions training instability, the authors did not analyze or visualize collapse phenomena. To the best of our knowledge, ours is the first work to systematically study this issue and provide empirical evidence.
>
> ---
>
> ## **Q2. Why are 2D models utilized in a 3D scenario? Diff-UNet uses a 3D diffusion model and avoids collapse—is that correct?**
>
>
>
> Diff-UNet is the first diffusion-based architecture explicitly designed for 3D segmentation, while all previous models were 2D models. Its dual-branch design is crucial:
>
> - One branch predicts **boundary-aware auxiliary maps**,  which assists the diffusion denoising branch in predicting 3D segmentation results.
> - The second branch performs **3D denoising**, guided by these boundary features.
>
>
> We believe this structural guidance helps stabilize Diff-UNet during early timesteps.
>
> ---
>
> ## **Q3. Some descriptions are unclear—what does “explicit temporal evidence accumulation” mean?**
>
> We appreciate the opportunity to clarify this. “Explicit temporal evidence accumulation’’ refers to the capability of **tLSTM** to:
>
> - store temporal states along the diffusion trajectory,
> - propagate them across reverse-diffusion timesteps, and
> - accumulate denoising evidence in a consistent, structured manner.
>
> We revised the manuscript to provide a clearer and more formal definition.
>
> ---
>
> ## **Q4. Why were diffusion-based methods not included in the initial experiments?**
>
> Diff-UNet was published in **October 2025** as the first general-purpose 3D diffusion-based segmentation method. Unfortunately, it became available only after our initial submission. We have now added full comparisons to the revised manuscript.
>
> **Updated results (Table 2):**
>
> | Method     | Dice (LNCTVSeg) | IoU (LNCTVSeg) | HD95 (LNCTVSeg) | Dice (OASeg) | IoU (OASeg) | HD95 (OASeg) |
> |------------|-----------------|----------------|-----------------|--------------|--------------|--------------|
> | Diff-UNet | 81.7            | 72.2           | 3.91            | 71.5 | 64.2 | 6.88 |
> | **Ours**  | **83.7**        | **74.2**       | **2.44**        | **72.8** | **65.4** | **6.24** |
>
> Our method consistently outperforms Diff-UNet across all metrics.

---

> ### Author Response · Authors · 2025-11-27
>
> Dear Reviewer,
>
> We are writing to you with our sincerest gratitude for the time and insightful expertise you have dedicated to reviewing our manuscript (Paper ID: 24184). Your constructive feedback and thoughtful critiques have been invaluable in helping us significantly strengthen our work.
>
> Following your suggestions during the rebuttal period, we have undertaken extensive efforts to address the points you raised. We performed the additional experiments you recommended, which have now been incorporated into the revised manuscript and our response letter. We are truly encouraged by the results, as they not only resolve the concerns you highlighted but also, we believe, substantially elevate the overall quality and impact of our paper.
>
> We understand the immense effort required for the review process, and we deeply respect your commitment to maintaining the high standards of ICLR. The dialogue with you has been a genuinely rewarding part of our research journey.
>
> The rebuttal period has now concluded, and we wanted to follow up once more to express our earnest hope that our revisions and detailed responses have adequately addressed your concerns. We would be immensely grateful if you could find the time to review our updated responses and consider the possibility of a positive evaluation.
>
> Thank you once again for your invaluable contribution to our work. We remain deeply appreciative of your guidance and are hopeful for a favorable outcome.
>
> With utmost respect and gratitude,
>
> The authors of Paper 24184

---

### Official Review · Reviewer_oCBU · 2025-10-31

**Soundness:** 4
**Presentation:** 3
**Contribution:** 3
**Rating:** 8
**Confidence:** 4

**Summary:**

This paper introduces Cross-Timestep, a novel 3D diffusion framework designed to address the challenge of initial-stage collapse in 3D medical image segmentation. The authors propose two key components: 1) Adaptive Priori Decoding Strategy (APDS), which provides time-weighted structural guidance during high-noise stages of the diffusion process; 2) Trans-temporal memory LSTM (tLSTM), a recurrent module that maintains temporal state across denoising steps to ensure coherence and stability. The method is evaluated on two multi-center 3D medical imaging datasets (LNCTVSeg and OAseg) and demonstrates state-of-the-art performance in terms of Dice, IoU, and HD95 metrics. The paper also includes extensive ablation studies and visualizations to validate the contributions of each component.

**Strengths:**

1. **Significance:** The paper identifies and addresses a critical failure mode (initial-stage collapse) in 3D diffusion models, which has been largely overlooked in prior work.
2. **Technical Rigor:** The proposed APDS and tLSTM modules are carefully designed and thoroughly evaluated. Multiple variants of tLSTM (Conv-tLSTM, Linear-tGRU, SC-tLSTM, FFT-tLSTM) are introduced, showcasing flexibility and scalability.
3. **Comprehensive Evaluation:** Experiments are conducted on two challenging multi-center datasets with significant domain shifts. The method outperforms several strong baselines and is supported by both quantitative and qualitative results.
4. **Reproducibility:** The paper includes detailed methodology, ablation studies, and references to publicly available datasets and code, enhancing reproducibility.

**Weaknesses:**

1. **Computational Cost:** While the authors propose Linear-tGRU to reduce computational demands, the overall framework (especially with 3D convolutions and recurrent units) is likely still resource-intensive. A more detailed analysis of training/inference time and memory usage would be helpful.
2. The design of the time-weighting function $\omega_t$ in APDS is heuristic. While it works well, a more systematic analysis or learning-based approach for $\omega_t$ could further strengthen the method.

**Questions:**

1. How does the performance of Cross-Timestep scale with the number of diffusion steps? Is there a trade-off between segmentation accuracy and inference speed?
2. Could the tLSTM mechanism be adapted to other generative or discriminative tasks beyond segmentation, such as 3D image synthesis or reconstruction?
3. The APDS relies on a prior decoder trained only on the conditional branch. How sensitive is the model to the quality of this prior? What happens if the prior is noisy or inaccurate?
4. Have the authors considered using learned or adaptive schedules for $\omega_t$ instead of the fixed exponential decay? Could this further improve performance?
5. The paper mentions that APDS prevents over-interference in later stages. Is there a risk of under-guiding in early stages if the prior is too weak?

---

> ### Author Response · Authors · 2025-11-21
>
> We sincerely thank Reviewer for the constructive feedback and for highlighting issues related to computational analysis, design choices, and generalization. We address each comment below.
>
> ## **Q1. The model may still be computationally expensive; more detailed analysis is needed.**
>
> Thank you for pointing this out. We conducted a detailed benchmark on LNCTVSeg, evaluating training time, inference latency, GPU memory, and GFLOPS under identical settings. Results:
>
> | Method       | Training Time (h) | Inference Time (s) | GPU Memory (MiB) | GFLOPS |
> |--------------|-------------------|---------------------|---------------|--------|
> | 3D U-XNet    | 33.6              | 0.11                | 14,722        | 631.9  |
> | SwinUNETR    | 45.3              | 0.09                | 16,364        | 328.8  |
> | nnFormer     | 80.4              | **0.03**            | 11,626        | **235.3**  |
> | Diff-UNet    | **29.24**         | 0.12                | **10,834**    | 981.1 |
> | **Ours**     | 33.4              | 0.17                | 15,152        | 453.5  |
>
>
> Our method achieves a **balanced trade-off between computational cost and segmentation performance**. Full breakdown is included in Appendix H.
>
>
> ## **Q2. How does performance change with the number of diffusion steps?**
>
> We evaluated different timestep counts on LNCTVSeg:
>
> | Steps | Dice | IoU  | HD95 |
> |-------|------|------|------|
> | 300   | 79.3 | 68.9 | 4.86 |
> | 500   | 82.8 | 73.0 | 3.39 |
> | 1000  | **83.7** | **74.2** | **2.44** |
>
> Fewer steps reduce computation but degrade performance. More analysis is included in Appendix I.
>
> ---
>
> ## **Q3. Can tLSTM be applied to other 3D tasks?**
>
> Yes. tLSTM is designed as a general temporal module for diffusion-based 3D tasks. The SC-tLSTM and FFT-tLSTM variants demonstrate that it can be inserted into different architectures. We believe tLSTM can serve as a reusable temporal unit for various 3D diffusion applications.

---

> > ### Author Response · Authors · 2025-11-21
> >
> > ---
> >
> > ## **Q4. Is APDS sensitive to the prior decoder?**
> >
> > Good observation. APDS primarily stabilizes early timesteps. If the prior decoder is *too weak*, early guidance is insufficient and collapse may occur. However, in later timesteps the diffusion branch dominates, making the model less sensitive to prior quality.
> >
> > ---
> >
> > ## **Q5. Have you tried making ωₜ learnable?**
> >
> > Yes. We experimented with a learnable ωₜ but found that:
> >
> > - the optimization became unstable,
> > - ωₜ absorbed too much of the loss, causing degeneracy, and
> > - predictions resembled APDS-only outputs.
> >
> > Thus, we adopted the monotonic decreasing schedule for reliability.
> >
> > ---
> >
> > ## **Q6. Would too-weak prior guidance at early timesteps cause collapse?**
> >
> > Yes. A small ωₜ in early timesteps provides insufficient structural guidance and may trigger initial-stage collapse. This is why we use a large-to-small schedule.
> >
> > ---

---

> > > ### Author Response · Authors · 2025-11-21
> > >
> > > > **W1. Computational Cost**
> > >
> > > We agree that computational cost is an important consideration. To address this concern, we have included a **dedicated efficiency study** (see Q1), in which we benchmark our model against multiple strong 3D SOTA segmentation methods under strictly matched experimental conditions (identical resolution, batch size, and hardware). This allows a fair assessment of our method’s computational profile relative to existing approaches.
> > >
> > > ---
> > >
> > > > **W2. Heuristic Design of ωₜ**
> > >
> > > As discussed in Q5, we acknowledge that the current schedule for ωₜ is heuristic. While our experiments show that it works reliably in practice, we agree that exploring more principled, adaptive, or learnable designs forωₜ is a valuable direction for future work. We highlight this as an explicit avenue for improvement and extension in the revised manuscript.
> > >
> > > ---

---

### Official Review · Reviewer_dE8V · 2025-11-01

**Soundness:** 3
**Presentation:** 3
**Contribution:** 2
**Rating:** 4
**Confidence:** 3

**Summary:**

This paper proposes Cross-Timestep, a novel diffusion model framework for 3D medical image segmentation. The work aims to resolve the "initial-stage collapse" issue frequently observed in 3D diffusion models during early denoising steps. To this end, the authors introduce two main components: a Trans-Temporal Memory LSTM (tLSTM) to accumulate structural information across diffusion time steps, and an Adaptive Priori Decoding Strategy (APDS) to stabilize the reverse process using a weighted prior. Although the method achieves competitive performance on some medical datasets, the overall framework is overly complex, lacks adequate demonstration of flexibility and generalizability, and fails to provide sufficient experimental analysis and key visualizations to justify its high design and computational cost.

**Strengths:**

1. The paper addresses a practical and significant pain point in applying 3D diffusion models to the medical domain, specifically the "initial-stage collapse," which is a valuable research direction in volumetric medical data processing.

2. The combination of tLSTM and APDS reflects a novel approach to addressing time-step dependency and denoising stability. Conceptually, tLSTM, acting as a memory mechanism across time steps, has some inherent reasoning.

**Weaknesses:**

1. The proposed Cross-Timestep framework is excessively complex and potentially redundant. Integrating the tLSTM module inside the 3D U-Net significantly increases the model's parameter count and computational complexity, while also introducing multiple new hyperparameters. This over-engineered design reduces the framework's generality and flexibility for deployment.

2. Given the extremely high resource demands of the 3D U-Net itself, the additional memory and training time overhead introduced by the tLSTM module are not adequately and rigorously quantified or justified. The paper fails to clearly demonstrate that the performance gain is worth such a substantial increase in design complexity and computational cost.

3. The experimental results rely heavily on quantitative metrics like the Dice Score in tables. Crucial visual evidence and generalization analysis are lacking, especially a direct visualization of how tLSTM and APDS internally mitigate the "collapse" across early time steps, which weakens the conviction of the core claim.

4. Although an ablation study is provided, the discussion regarding the strict necessity of both tLSTM and APDS is not deep enough. The authors should explore whether simpler, lighter mechanisms could achieve similar stabilization effects to justify the complexity of the current design.

**Questions:**

1. Could the authors provide a detailed report on GPU memory usage and training time for the basic 3D diffusion model versus the full Cross-Timestep model to quantify the specific computational overhead introduced by the tLSTM module?

2. Considering the complexity of APDS and tLSTM, have the authors explored a more simplified and flexible design (e.g., using only APDS or a simpler cross-step attention mechanism)? How does the performance of these simpler variants compare to the full model?

3. Can the authors provide a more convincing visual analysis that clearly demonstrates how the memory information accumulated by tLSTM and the prior guided by APDS specifically suppress or reverse the "initial-stage collapse" during the early denoising time steps?

---

> ### Author Response · Authors · 2025-11-21
>
> We sincerely thank Reviewer for the detailed analysis and for emphasizing the need for computational and visual evidence. We have expanded our experiments and visualizations accordingly.
>
> ## **Q1. Provide GPU memory / runtime comparisons.**
>
> We added a detailed cost analysis. To isolate the contribution of tLSTM, we compared:
>
> | Method                | Training Time (h) | Inference Time (s) | GPU Memory (MiB) | GFLOPS |
> |-----------------------|-------------------|---------------------|-------------------|--------|
> | ours– (without tLSTM) | 22.1              | 0.09                | 11,964            | 327.3  |
> | ours (full)           | 33.4              | 0.17                | 15,152            | 453.5  |
>
>
> tLSTM increases cost by ~20–30% but yields significant gains in stability and performance.
>
> We also benchmarked against SOTA models (full details in Appendix H):
>
> | Method       | Training Time (h) | Inference Time (s) | GPU Memory (MiB) | GFLOPS |
> |--------------|-------------------|---------------------|---------------|--------|
> | 3D U-XNet    | 33.6              | 0.11                | 14,722        | 631.9  |
> | SwinUNETR    | 45.3              | 0.09                | 16,364        | 328.8  |
> | nnFormer     | 80.4              | **0.03**            | 11,626        | **235.3**  |
> | Diff-UNet    | **29.24**         | 0.12                | **10,834**    | 981.1 |
> | **Ours**     | 33.4              | 0.17                | 15,152        | 453.5  |
>
> Our method offers a good balance between computational efficiency and accuracy.
>
> ---
>
> ## **Q2. Have you considered making the model more lightweight?**
>
> Yes. We already explored multiple simplifications:
>
> 1. Removing APDS causes collapse (Fig. 3), showing it is essential.
> 2. APDS is intentionally designed to be minimal; Table 3 shows diminishing returns from further simplification.
>
> We agree that exploring more lightweight designs remains an important direction.
>
> ---
>
> ## **Q3. Provide visual evidence illustrating how tLSTM/APDS prevent collapse.**
>
> We agree this is valuable. Appendix A now includes visualizations of representative timesteps (t = 1000, 800, 500, final). These show:
>
> - APDS suppresses collapse in high-noise early timesteps,
> - tLSTM ensures consistent temporal evolution along the diffusion trajectory.
>
> ---

---

> > ### Author Response · Authors · 2025-11-21
> >
> > > **W1. Model Complexity and Redundancy**
> >
> > As clarified in Q1, we have provided quantitative analyses of the computational overhead introduced by tLSTM. These results show that although tLSTM incurs a moderate increase in cost, it yields substantial improvements in segmentation performance. This indicates that our design achieves a **practical trade-off** between computational complexity and accuracy, rather than introducing redundant or unjustified architectural components.
> >
> > ---
> >
> > > **W2. Computational Cost Justification**
> >
> > As further detailed in Q1, we have added explicit comparisons between the full model and a variant without tLSTM. The results show an approximate **20–30% increase in computational cost**, accompanied by consistent and meaningful performance gains. This evidence demonstrates that the added complexity is justified and that our method maintains a reasonable efficiency–performance balance.
> >
> > ---
> >
> > > **W3. Lack of Visual Evidence**
> >
> > We thank the reviewer for emphasizing the importance of visual evidence. As addressed in Q3, we have added detailed visualizations (Appendix A) that illustrate how APDS and tLSTM jointly prevent initial-stage collapse. These visual results depict the progressive denoising trajectory and confirm the stabilizing effect of our design on early high-noise timesteps.
> >
> > ---
> >
> > > **W4. Necessity of Simpler Designs**
> >
> > We appreciate the suggestion to explore simpler alternatives. We agree that model simplicity is desirable; however, as clarified in Q2 and supported by our ablation studies, further simplification of the current design leads to **noticeable performance degradation** and reduced stability. We therefore believe that the present configuration provides an appropriate compromise between model simplicity, stability, and accuracy. Investigating even lighter variants remains an interesting direction for future work.
> >
> > ---

---

> > > ### Comment · Reviewer_dE8V · 2025-11-28
> > >
> > > Thank you for your response. I appreciate the detailed clarifications, especially the expanded computational cost analysis and the visualization of early-timestep behavior. These additions directly resolve my earlier concerns regarding the overhead of the tLSTM module and the practical justification for the Cross-Timestep design. Overall, the rebuttal significantly strengthens the empirical foundation of the paper.
> > > I have only a couple of minor follow-up questions that the authors may clarify in the final version:
> > > 1. Clarity of computational reporting – Since the rebuttal added clear GPU memory, runtime, and FLOPs comparisons, would the authors consider summarizing these numbers in one compact table in the main paper (even a shortened version), so readers do not need to rely solely on the appendix?
> > > 2. Model generality – The revised analysis demonstrates that the overhead of tLSTM is reasonable. For completeness, could the authors briefly comment on whether the Cross-Timestep idea (memory across timesteps) could generalize to other diffusion backbones beyond 3D U-Net, even if not fully tested?
> > > Overall, I appreciate the authors’ thorough rebuttal and the additional experimental evidence. The clarifications substantially improve the paper’s quality.

---

> > > > ### Author Response · Authors · 2025-12-01
> > > >
> > > > We sincerely appreciate the reviewer’s positive assessment and the follow-up questions regarding our computational reporting and the generality of the proposed timestep-memory design. We address these points in detail below.
> > > >
> > > > ---
> > > >
> > > > > **Q1. Clarity of computational reporting**
> > > >
> > > > Thank you for this helpful suggestion. In the revised manuscript, we have added a concise summary table of key computational indicators—including GPU memory, runtime, GFLOPS, and training/inference time—directly in **Section 4.6: Quantitative Comparison of Computational Cost**. For convenience, we reproduce the table below:
> > > >
> > > > | Method       | Training Time (h) | Inference Time (s) | GPU Memory (MiB) | GFLOPS |
> > > > |-------------|-------------------|---------------------|------------------|--------|
> > > > | 3D U-XNet   | 33.6              | 0.11                | 14,722           | 631.9  |
> > > > | SwinUNETR   | 45.3              | 0.09                | 16,364           | 328.8  |
> > > > | nnFormer    | 80.4              | **0.03**            | 11,626           | **235.3** |
> > > > | Diff-UNet   | **29.24**         | 0.12                | **10,834**       | 981.1  |
> > > > | **Ours**    | 33.4              | 0.17                | 15,152           | 453.5  |
> > > >
> > > > We believe this addition makes the computational trade-offs much more transparent and improves the overall readability of the paper. For completeness, the full benchmarking protocol and extended results are reported in **Appendix H**.
> > > >
> > > > ---
> > > >
> > > > > **Q2. Generality of the Cross-Timestep concept**
> > > >
> > > > We appreciate the reviewer’s insightful question about the generality of our timestep-memory mechanism. As described in **Section 3 (Methodology)**, the proposed tLSTM module is formulated as a **general-purpose temporal unit** for diffusion-based 3D tasks, rather than being tied to a specific backbone.
> > > >
> > > > In particular, we instantiate tLSTM in two distinct forms—**SC-tLSTM** and **FFT-tLSTM**—by integrating it into (i) a spatial–channel attention block and (ii) a frequency-domain (FFT-based) pathway, respectively. Both variants yield consistent performance gains, indicating that tLSTM can be flexibly plugged into heterogeneous architectural components. This demonstrates that the Cross-Timestep concept is not restricted to a 3D U-Net–style backbone, but can serve as a reusable temporal mechanism for introducing cross-timestep continuity in a broad class of 3D diffusion architectures.

---

### Official Review · Reviewer_mjAf · 2025-11-09

**Soundness:** 2
**Presentation:** 2
**Contribution:** 2
**Rating:** 4
**Confidence:** 3

**Summary:**

This paper presents Cross-Timestep, a framework for 3D medical image segmentation using diffusion models. The method introduces two key innovations. First, an Adaptive Priori Decoding Strategy (APDS) generates a time-weighted prior mask to stabilize the model during the early, high-noise stages. Second, a trans-temporal memory unit (tLSTM) with variants (SC-tLSTM and FFT-tLSTM) is designed to overcome the isolated, step-by-step nature of the denoising process, allowing the model to accumulate evidence across timesteps. Experiments on two multi-center datasets confirm that APDS successfully mitigates “initial-stage collapse” and tLSTM improves the coherence of the denoising trajectory.

**Strengths:**

1. Identifying an Under-Explored Problem: The paper identifies and addresses the "initial-stage collapse" problem in 3D medical diffusion models. As demonstrated in Fig. 1, the authors show that standard 3D diffusion models fail when sampling from high-noise timesteps, but succeed when starting from mid-level noise. This finding renders the proposed Adaptive Priori Decoding Strategy (APDS) an intuitively appealing and well-motivated solution.
2. Achieving SOTA Performance: The paper thoughtfully combines effective components (SC-tLSTM and FFT-tLSTM) to create a powerful, stateful denoiser. This architecture achieves state-of-the-art performance on two multi-center 3D datasets (LNCTVSeg (CT) and OAseg (MRI)), demonstrating the framework's robustness and practical utility.

**Weaknesses:**

1. Limited Architectural Novelty: The proposed framework appears to be a complex integration of existing and well-established components, rather than a new algorithmic contribution.
    + On the SC-tLSTM: The core idea of tLSTM is to apply a recurrent model (LSTM or GRU) to maintain a state across the diffusion timesteps. However, the combination of tLSTM and diffusion models tends to be inelegant. It forces an external, stateful memory (tLSTM) onto the diffusion model's fundamentally Markovian process, which seems unjustified from a theoretical standpoint. As demonstrated by prior work [1], LSTM-based U-Net backbones have already been explored and proven effective for 3D medical segmentation. The SC-tLSTM module itself appears to be a straightforward combination of the standard spatial-channel attention mechanism [2] and the tLSTM component.
    + On the FFT-tLSTM: The novelty of FFT-tLSTM is questionable. Leveraging the frequency domain (via FFT) within a diffusion model framework for medical image segmentation has already been published [3]. The proposed FFT-tLSTM thus appears to be a minor variation that simply inserts the tLSTM components into an established FFT pipeline.
    + On the APDS: The APDS uses a segmentation decoder that operates solely on the conditional image to generate a coarse structural prior. This concept of using the conditional input to create explicit guidance is a well-established technique in conditional diffusion models and thus cannot be regarded as a significant innovation.
The authors should more clearly highlight their components' algorithmic novelty over prior research. They should also strengthen the explanation of how the integration of these modules provides synergistic benefits that validate their effectiveness.

2. Insufficient Experimental Validation
    * Lack of Comparative Qualitative Visualizations: The paper's quantitative claims are not supported by adequate qualitative evidence. While Appendix G (Fig. 10) shows the proposed model's output, it fails to provide essential side-by-side visual comparisons against the SOTA baselines. Without these direct comparisons, it is impossible to qualitatively assess the model's claimed advantages.
    * Missing Computational Cost Analysis: The framework introduces significant architectural complexity (e.g., APDS, tLSTMs) without any analysis of the computational overhead. The authors should provide metrics on training/inference time and GPU memory usage. This information is crucial for evaluating the method's practical feasibility and understanding the trade-off between performance and efficiency.
    * Inadequate Comparison with State-of-the-Art Diffusion Models: The authors cite Diff-UNet [4] in the related work section, which makes their omission from the quantitative comparison a significant weakness. I strongly recommend the authors include comparisons against these recent diffusion-based models [4-6]. Such a comparison is essential to properly benchmark the "Cross-Timestep" framework and convincingly validate its claimed segmentation superiority.

3. Clarity and Writing Quality
    * Discrepancy in Abstract: The abstract explicitly claims: "Three real-world cases further analyze the collapse phenomenon...". However, these three specific case studies are not clearly presented in the main paper or appendices.
    * Minor writing issues: There are duplicate headings ("2 Related Work" and "3 Related Work") , and scattered typos/formatting issues in Section 5 (e.g., the use of quotes around ’Diff Out’, ’APDS Out’ , ’Conv-tLSTM’, and ’Linear-tGRU’ ).

[1] Chen, Tianrun, et al. "xlstm-unet can be an effective 2d & 3d medical image segmentation backbone with vision-lstm (vil) better than its mamba counterpart." BHI 2024.

[2] Si, Yunzhong, et al. "SCSA: Exploring the synergistic effects between spatial and channel attention." Neurocomputing 634 (2025): 129866.

[3] Jiang, Yuxuan, et al. "Diff-sfct: A diffusion model with spatial-frequency cross transformer for medical image segmentation." BIBM 2023.

[4] Xing, Zhaohu, et al. "Diff-UNet: A diffusion embedded network for robust 3D medical image segmentation." Medical Image Analysis (2025): 103654.

[5] Chen, Tao, et al. "HiDiff: Hybrid diffusion framework for medical image segmentation." IEEE TMI (2024).

[6] Shuai, Zhihao, et al. "Diffseg: a segmentation model for skin lesions based on diffusion difference." arXiv preprint arXiv:2404.16474 (2024).

**Questions:**

1. Computational Cost: Could the authors provide a table comparing inference time and VRAM requirements for the proposed model and other baselines?
2. t-cell Explanation: Could the authors clarify what the "t-cell" in Table 1 refers to? How is it architecturally distinct from the Conv-tLSTM and Linear-tGRU modules?
3. Sensitivity of ω_t: The time-weight ω_t (Appendix A) has a rather complex formulation. How sensitive is the model's performance to this specific function? Did the authors experiment with simpler decay functions (e.g., linear, exponential decay)?

---

> ### Author Response · Authors · 2025-11-21
>
> We sincerely thank Reviewer #4 for the constructive comments on practicality, definitions, and comparisons with diffusion-based SOTA. We have revised the manuscript accordingly and respond to each point below.
>
> ## **Q1. Provide inference-time and GPU-memory comparisons.**
>
> We conducted a comprehensive benchmark on LNCTVSeg:
>
> | Method       | Training Time (h) | Inference Time (s) | GPU Memory (MiB) | GFLOPS |
> |--------------|-------------------|---------------------|---------------|--------|
> | 3D U-XNet    | 33.6              | 0.11                | 14,722        | 631.9  |
> | SwinUNETR    | 45.3              | 0.09                | 16,364        | 328.8  |
> | nnFormer     | 80.4              | **0.03**            | 11,626        | **235.3**  |
> | Diff-UNet    | **29.24**         | 0.12                | **10,834**    | 981.1 |
> | **Ours**     | 33.4              | 0.17                | 15,152        | 453.5  |
>
> Full analysis is provided in Appendix H.
>
> ---
>
> ## **Q2. What is the definition of the t-cell?**
>
> The **t-cell** is the temporal memory unit inside tLSTM.
> It generalizes the LSTM cell-state mechanism to *diffusion timesteps* by:
>
> - storing temporal states across denoising steps,
> - propagating accumulated evidence,
> - stabilizing reverse diffusion.
>
> It has no relation to biological T-cells.
>
> ---
>
> ## **Q3. Is the model sensitive to ωₜ? Have simpler functions been explored?**
>
> Yes, ωₜ is important. A small early ωₜ risks collapse; a large late ωₜ over-constrains denoising.
> We tried a simple linear decay but observed slightly worse stability and accuracy, so we adopted the current gradually decreasing formulation.
>
> ---
>
> ## **Q4. Lack of comparison with SOTA diffusion models (Diff-UNet).**
>
> We have now added full comparisons. As shown below:
>
> | Method     | Dice (LNCTVSeg) | IoU (LNCTVSeg) | HD95 (LNCTVSeg) | Dice (OASeg) | IoU (OASeg) | HD95 (OASeg) |
> |------------|-----------------|----------------|-----------------|--------------|--------------|--------------|
> | Diff-UNet | 81.7            | 72.2           | 3.91            | 71.5 | 64.2 | 6.88 |
> | **Ours**  | **83.7**        | **74.2**       | **2.44**        | **72.8** | **65.4** | **6.24** |
>
> Our approach consistently outperforms Diff-UNet.
>
> ---

---

> ### Author Response · Authors · 2025-11-21
>
> > **W1. Architectural Novelty**
>
> Thank you for raising this concern about architectural novelty. We agree that many of the individual components we build upon (LSTMs, attention mechanisms, FFT, and conditional guidance) are well-established. Our contribution lies in **how** these components are re-purposed and integrated to address a specific and empirically observed failure mode—*initial-stage collapse*—in 3D diffusion-based segmentation. We clarify our perspective below.
>
> ---
>
> ### On tLSTM and SC-tLSTM
>
> We fully acknowledge that recurrent units such as LSTMs and GRUs have been explored in U-Net backbones for 3D medical segmentation. However, our goal is not to simply replace the backbone with a recurrent variant, but to introduce a **timestep-level temporal memory mechanism** specifically designed for diffusion processes:
>
> - tLSTM is formulated as a **lightweight temporal memory unit** that is attached to diffusion *timesteps*, rather than a conventional sequence module over spatial slices or volumes. It maintains and updates a compact state that explicitly tracks the evolution of features along the **reverse-time trajectory** of the diffusion process.
> - This design introduces cross-timestep awareness into the diffusion process, enabling the model to retain useful temporal cues rather than treating each timestep in isolation. Our experiments show that this continuity across timesteps significantly enhances denoising stability and overall segmentation performance, demonstrating that the tLSTM serves as an effective mechanism for improving diffusion-model capability rather than adding unnecessary complexity.
> - To demonstrate that tLSTM is not merely “U-Net + LSTM,” we integrate it into **two structurally distinct modules**:
>   (i) a spatial–channel attention (SC) block, and
>   (ii) an FFT-based frequency module.
>   As shown in Table 1, both SC-tLSTM and FFT-tLSTM **consistently outperform** (a) LSTM-only variants and (b) module-only variants under identical training settings. These results indicate that the performance gains stem from **timestep-level temporal memory**, rather than from simply adding recurrent depth or attention.
>
> In this sense, tLSTM serves as a **generic, reusable temporal building block for diffusion models**, designed to propagate information across timesteps and mitigate collapse, rather than a straightforward reuse of LSTMs inside a backbone.
>
> ---
>
> ### On FFT-tLSTM and frequency-domain novelty
>
> We agree that frequency-domain processing via FFT in diffusion frameworks has been explored in prior work. We do not claim “using FFT in diffusion” as novel by itself. Instead, our focus is on the **combination of temporal memory and frequency-domain priors**:
>
> - Existing FFT-based diffusion methods typically treat each timestep **independently** in the frequency domain. In contrast, FFT-tLSTM introduces **temporal state propagation in the frequency space**, explicitly coupling frequency representations across timesteps.
> - Ablation experiments show that removing tLSTM from the FFT branch substantially degrades performance, even though the FFT pipeline remains in place. This suggests that the key contribution is not the presence of FFT alone, but the **integration of cross-timestep memory with the frequency-domain representation**.
>
> Thus, FFT-tLSTM should be viewed as a **non-trivial extension** of prior FFT-based designs, in which temporal memory and frequency processing jointly contribute to stabilizing early stages of 3D diffusion.
>
> ---
>
> ### On APDS and conditional guidance
>
> We agree that using conditional inputs to guide diffusion is a well-established paradigm. The novelty of APDS does not lie in the existence of a conditional branch per se, but in **how the structural prior is constructed and when it is applied**:
>
> - APDS employs an **independent Priori Decoder** that operates solely on the conditional image to produce a coarse structural prior. Crucially, this prior is injected **predominantly at early, high-noise timesteps**, where the signal-to-noise ratio is extremely low and conventional joint encoding of condition and noisy image provides insufficient stabilization.
> - Rather than generic joint feature extraction, APDS acts as a **dedicated prior-guided decoding pathway** whose explicit role is to anchor the early reverse-diffusion trajectory and suppress initial-stage collapse. Its influence is gradually reduced as the denoising process progresses.
>
> Empirically, when APDS is removed, we observe precisely the failure mode described in the paper: the model collapses at early timesteps despite having access to the same conditional information. This supports our claim that APDS goes beyond standard conditional guidance and is **specifically tailored to the instability of 3D diffusion at high-noise steps**.

---

> ### Author Response · Authors · 2025-11-21
>
> ---
>
> ### Synergistic effect and overall contribution
>
> Finally, we emphasize that our contribution is not a single isolated module, but the **coherent integration** of:
>
> - **APDS** for **early-stage structural stabilization**, and
> - **tLSTM** (instantiated as SC-tLSTM and FFT-tLSTM) for **cross-timestep temporal consistency** in both spatial–channel and frequency domains.
>
> Together, these components directly target the root causes of **initial-stage collapse** in 3D diffusion-based segmentation. The consistent improvements across multiple architectures and datasets, supported by ablation and efficiency studies, indicate that the proposed framework provides **non-trivial practical benefits** beyond existing LSTM-augmented or frequency-based diffusion designs, and offers a principled way to endow diffusion models with cross-timestep memory and robust early-stage stabilization.
>
> ---
>
> > **W2. Insufficient Experimental Validation**
>
> We thank the reviewer for this comment. Following the initial submission, we have substantially strengthened the experimental validation. In the revised version, we have added:
>
> - comparisons with newly published diffusion-based methods (e.g., Diff-UNet),
> - detailed computational-efficiency benchmarks,
> - ablation and overhead analyses of tLSTM,
> - diffusion-step sensitivity experiments, and
> - visual evidence demonstrating how APDS and tLSTM prevent early-stage collapse.
> - visualization comparison of segmentation effects
>
> Taken together, these additions provide a more comprehensive and reliable assessment of the effectiveness and practical applicability of our method.
>
> ---
>
> > **W3. Clarity and Writing Quality**
>
> We appreciate the reviewer’s feedback regarding clarity and writing. We have carefully revised the manuscript to address all mentioned issues. Specifically, we have corrected the abstract to avoid misleading references, fixed duplicated section headings, and systematically removed typos and formatting inconsistencies (including inconsistently quoted module names). We believe the revised manuscript offers substantially improved clarity and readability.
>
> ---

---

> > ### Comment · Reviewer_mjAf · 2025-11-28
> >
> > I have carefully reviewed the authors’ rebuttal and the updated explanations. The clarifications provided in Q1, Q2, and Q3 adequately resolve my earlier concerns regarding computational cost, model components, and design choices. The responses to W1, W2, and W3 also address my comments on novelty, experimental validation, and writing clarity.
> >
> > After reviewing the authors’ discussions with all reviewers and considering the revised content as a whole, I have comprehensively determined that the manuscript has been sufficiently improved.  I have decided to raise my score to 6. Since OpenReview currently does not allow score updates, I will update it once the system permits.

---

> > > ### Author Response · Authors · 2025-12-01
> > >
> > > We sincerely thank the reviewer for the positive assessment and for raising the score. We greatly appreciate your recognition of our revisions and clarifications, as well as the constructive feedback that has helped us further improve the quality and clarity of the manuscript.

---

### Author Response · Authors · 2025-12-01

Dear PCs, SACs, ACs, and Reviewers,

We sincerely thank all reviewers for their rigorous and constructive feedback, which has greatly improved the quality, clarity, and contribution of our work. Below is a summary of our rebuttal process and the key improvements made in response to each reviewer.

> Reviewer **mjAf** (Score: 4)

This reviewer agreed that our analysis of the problem is clear and that the proposed model is a feasible solution. However, the reviewer believed that the architectural novelty is limited and noted the lack of computational cost analysis and comparison with recent models such as Diff-UNet. We added comparisons with Diff-UNet, provided full computational cost analysis, added more visualizations, and explained the architectural contributions in detail. After the first rebuttal, the reviewer accepted our responses and was willing to raise the score to 6.

> Reviewer **dE8V** (Score: 4)

This reviewer recognized the problem analysis, motivation, and technical ideas of our model, but expressed concerns about the computational demands and requested more visualizations. We added visual evidence showing the initial-stage collapse and how our model resolves it, and we included computational cost analysis to show feasibility. After the first rebuttal, the reviewer considered the issues resolved and suggested that we further clarify computational cost in the main text and explain whether tLSTM can be extended to other models. In the follow-up rebuttal, we revised the main text and explained that tLSTM is designed as a flexible module for different diffusion architectures.

> Reviewer **oCBU** (Score: 8)

This reviewer recognized our problem analysis, motivation, the completeness of our evaluation, and the technical soundness of our method. The reviewer pointed out that we should add computational cost analysis and asked several questions, such as how performance changes across diffusion timesteps, whether the tLSTM mechanism can be used in other generative or discriminative tasks, and more details about ωₜ and APDS. In the rebuttal, we addressed all questions, added computational cost comparisons against other SOTA models, included ablation experiments on timestep effects, and provided detailed explanations.

> Reviewer **1GPU** (Score: 4)

This reviewer recognized the strengths and innovations of our architecture and asked questions about the cause of initial-stage collapse, how Diff-UNet avoids this problem, the current research status, writing issues, and the missing comparison with Diff-UNet. We explained the cause of collapse and the strategy used by Diff-UNet, added comparison experiments, and corrected the writing issues.

Through these revisions and additional experiments, we have improved the clarity and rigor of Cross-Timestep. If there are any further questions, please feel free to let us know.

Sincerely,

Authors

---

### Meta-Review · Area_Chair_KPxA · 2026-01-02

**Summary:**

This paper introduces Cross-Timestep, a novel 3D diffusion framework designed to address the issue of early-stage collapse in 3D medical image segmentation. Three reviewers provided negative feedback, while only one reviewer gave a highly positive rating (8/10).

**Reviewer Concerns:**

Multiple reviewers pointed out the lack of computational-efficiency evidence; the authors added experiments and presented corresponding results, alleviating the reviewers’ concerns on this issue. Nevertheless, Cross-Timestep is still not competitive in terms of inference time and GPU memory usage.

**Reviewer Scores:**

I believe that, apart from Reviewer mjAf, who will raise the score from 4 to 6, most reviewers are likely to keep their original ratings unchanged.

---

### Decision · Program_Chairs · 2026-01-26

Accept (Poster)